

# Water ages in the critical zone of long-term experimental sites in northern latitudes

Matthias Sprenger[1*], Doerthe Tetzlaff[2,3,1], Jim Buttle[4], Hjalmar Laudon[5], Chris Soulsby[1]

[1]Northern Rivers Institute, School of Geosciences, University of Aberdeen, Aberdeen, UK
(matthias.sprenger@abdn.ac.uk, c.soulsby@abdn.ac.uk)
[2]IGB Leibniz Institute of Freshwater Ecology and Inland Fisheries
(d.tetzlaff@igb-berlin.de)
[3]Humboldt University Berlin
[4]School of the Environment, Trent University, Ontario, Canada. (jbuttle@trentu.ca)
[5]Department of Forest Ecology and Management, Swedish University of Agricultural Sciences, Umeå, Sweden
(hjalmar.laudon@slu.se)

[*]*Correspondence to*: Doerthe Tetzlaff (d.tetzlaff@igb-berlin.de, +493064181661)

**Abstract**. As northern environments undergo intense respond due to a warming climate and altered land use practices, there is an urgent need for improved understanding of the impact of atmospheric forcing and vegetation on water storage and flux dynamics in the critical zone. We therefore assess the age dynamics of water stored in the upper 50 cm of soil, and in evaporation, transpiration or recharge fluxes at four soil-vegetation units of podzolic soils in the northern latitudes with either heather or tree vegetation (Bruntland Burn in Scotland, Dorset in Canada, and Krycklan in Sweden). We derived the age dynamics with the physically based SWIS (Soil Water Isotope Simulator) model, which has been successfully used to simulate the hydrometric and isotopic dynamics in the upper 50 cm of soils at the study sites. The modelled subsurface was divided into interacting fast and slow flow domains. We tracked each day's infiltrated water through the critical zone and derived forward median travel times (which show how long the water takes to leave the soil via evaporation, transpiration or recharge), and median water ages (to estimate the median age of water in soil storage or the evaporation, transpiration and recharge fluxes). Resulting median travel times were time-variant, mainly governed by major recharge events during snow melt in Dorset and Krycklan or during the wetter winter conditions in Bruntland Burn. Transpiration travel times were driven by the vegetation growth period with longest travel times (200 days) for waters infiltrated in early dormancy and shortest travel times during the vegetation period. However, long tails of the travel time distributions in evaporation and transpiration revealed that these fluxes comprised waters older than 100 days. At each study site, water ages of soil storage, evaporation, transpiration and recharge were all inversely related to the storage volume of the critical zone: water ages generally decreased exponentially with increasing soil water storage. During wet periods, young soil waters were more likely to be evapotranspired and recharged than during drier periods. While the water in the slow flow domain showed long-term seasonal dynamics and generally old water ages, the water ages of the fast flow domain were generally younger and much flashier. Our results provide new insights into the mixing and transport processes of soil water in the upper layer of the critical zone, which is relevant for hydrological modelling at the plot to catchment scales as the common assumption of a well-mixed system in the subsurface neither holds for the evaporation, transpiration nor recharge.



## 1 Introduction

Water ages are useful metrics to assess hydrological processes as they reveal interactions between storage and fluxes of water in a hydrological system (Hrachowitz et al., 2013; Tetzlaff et al., 2014; Pfister et al., 2017). Temporal variability of the water ages of streams results from the dynamics of hydro-meteorological conditions and wetness state of the catchment (Heidbüchel et al., 2013; Birkel et al., 2015; Hrachowitz et al., 2016). Thus, understanding the interplay between the climatic drivers and the state of the hydrological system is increasingly relevant in the light of climate and land use changes.

Northern environments have been shown to undergo particularly pronounced changes with an increase in land surface temperature (Hartmann et al., 2013) and tree cover (Forkel et al., 2016). Such altered climatic conditions and/or changes in vegetation cover are likely to modify the partitioning of water in the critical zone into evaporation, transpiration and recharge fluxes (Tetzlaff et al., 2013; Wang et al., 2018). As it has been shown that dynamics of evapotranspiration (ET) fluxes affect the water ages of catchment runoff (van der Velde et al., 2012; Birkel et al., 2012; Ali et al., 2014) and groundwater recharge (Sprenger et al., 2016), a better understanding of the ET dynamics and their effect on water ages and storage dynamics is needed. Water ages in ET fluxes also deserve increasing attention (Botter et al., 2011, 2010; Harman, 2015; Soulsby et al., 2016; van Huijgevoort et al., 2016) as the interlinkages between water stored in a hydrological system and the vegetation cover using that water are crucial to address challenges of water supply (Sterling et al., 2013; Wei et al., 2018).

Here, we address recent findings from water age theory, defined as an "inverse-storage effect" (Harman, 2015), where the storage in a hydrological system is related to the ages of the water fluxes leaving the system. At the catchment scale, several modelling studies have shown younger water ages during wet periods with high storage volumes (van der Velde et al., 2012; Benettin et al., 2013; Heidbüchel et al., 2013; Soulsby et al., 2015; Benettin et al., 2017). Berghuijs and Kirchner (2017) recently presented numerical experiments to investigate the relationship between catchment storage and runoff water ages, and detailed experimental work on a sloping lysimeter with pulse irrigation provided insights into the processes explaining the inverse relationship between storage and runoff age (Pangle et al., 2017). However, despite general acknowledgement that the conceptualization of the unsaturated zone in models affects runoff age estimates (McMillan et al., 2012; Heidbüchel et al., 2013; van der Velde et al., 2015), it is unclear if/how soils contribute to an inverse-storage effect. Given that soil storage as a percentage of total catchment storage was estimated to range between 20 % in the Scottish Highlands (Tetzlaff et al., 2014) to up to 80 % of the total catchment storage in rainfall-dominated alpine catchments (Staudinger et al., 2017), the role of soil water storage in water age dynamics needs to be more clearly identified. Further, an assessment of the variability of water ages within different pore spaces (e.g. mobile versus more tightly bound water - Brooks et al., 2010; Good et al., 2015; Sprenger et al., 2018; Smith et al., 2018) and with soil depth is still needed; this would also provide a test of the common assumption of a well-mixed system in tracer-aided modelling (van der Velde et al., 2015). Also, it has not yet been established how the ages of soil water storage and evaporation and transpiration fluxes are related to the variability of soil storage volumes.

To address these shortcomings, we examine the following questions in this paper: 1.) How long does it take for precipitation to leave the soil profile via evaporation, transpiration or recharge (travel times)? 2.) How old is the water





in these fluxes and the soil storage (water ages)? 3.) What are the controls on the dynamics of travel times and water ages in fluxes from, and storage within, the critical zone?

## 2    Methods

### 2.1    Study sites

The study sites were located in three long-term experimental catchments in the northern latitudes (map provided in Figure S 1): Bruntland Burn in the Scottish Highlands, UK (57°2' N 3°7' W), Dorset in south-central Ontario, Canada (45° 12' N 78° 49' W), and Krycklan in northern Sweden (64° 14' N 19° 46′ E). Climatic conditions range from temperate fully humid with cool summers at Bruntland Burn to cold fully humid with either warm summers in Dorset or cold summers in Krycklan. At all sites, there is a pronounced seasonality in air temperature and the long-term
annual means are 6.6 °C at Bruntland Burn, 4.8 °C at Dorset, and 1.8 °C at Krycklan. Precipitation is generally relatively evenly distributed over the year at all sites but snow accumulates at Dorset and Krycklan during the winter, leading to a pronounced soil infiltration pulse during snow melt in early spring. At Bruntland Burn, snow fall usually plays a minor role in the water balance (Ala-aho et al., 2017a), but rainfall occurs commonly at low intensities throughout the year (1000 mm yr$^{-1}$). Average annual precipitation is 1020 mm yr$^{-1}$ at Dorset and 622 mm yr$^{-1}$ at
Krycklan. A detailed comparison of the hydro-meteorological conditions at the three catchments was presented by Tetzlaff et al. (2015). Soils of the four sites were characterized as freely draining podzols of generally coarse texture ranging between loamy or silty sands to sand with an overlying organic layer of about 10-20 cm thickness. One site at Bruntland Burn was covered with Scots pine (*Pinus sylvestris*) and the other site was vegetated by Ericacae shrubs (*Calluna vulgaris*). Vegetation cover at Dorset was white pine (*Pinus strobus*) while at Krycklan, the soil was covered
by Scots pine (*Pinus sylvestris*). Rooting depths were observed in the field to be ~50 cm for the trees and 15 cm for the heather shrubs. Canopy coverage was about 60 % at the Bruntland Burn sites and about 89 % and 95 % at Dorset and Krycklan, respectively. All four locations were on hillslopes of low gradients (<9°). Detailed description of the soil and vegetation characteristics at the investigated sites were presented by Sprenger et al. (2018), where the sites were called "NF", "NH", "Pw", and "S22", respectively.

### 2.2    Data

Meteorological data including air temperature (°C), relative humidity (%), rainfall or snowmelt amount (mm d$^{-1}$) (Sprenger et al., 2018) and potential evapotranspiration (mm d$^{-1}$) estimated using the Penman-Monteith equation (Allen et al., 1998) were available on a daily basis at each catchment.
Soil hydraulic characteristics, as shown in the supplementary material (Figure S 2), were derived for Bruntland Burn
and Dorset from the pedotransfer functions provided by Schaap et al. (2001) using site specific soil textural and bulk density information (Sprenger et al., 2018). For Krycklan, the hydraulic parameters were estimated based on laboratory measurements (Nyberg et al., 2001).



## 2.3 Soil water flow and transport modelling

The simulations of travel times and water ages are based on tracking, in a 1-D soil hydraulic model, the infiltrated water (rainfall and snowmelt) with a virtual tracer in soil storage and fluxes leaving the soil. We applied the SWIS model as described in detail by Sprenger et al. (2018). The SWIS model solves the Richards equation for water flow and simulates tracer transport with the advection dispersion equation. The SWIS model can partition the subsurface into two flow domains (Figure S 3): a fast flow domain representing the soil pores that hold the water at pressure heads <600 hPa and a slow flow domain covering the pores with a water retention >600 hPa. A threshold of 600 hPa was chosen to divide the two pore domains, as this is approximately the pressure head applied by suction lysimeters to extract water. This definition allowed us to use stable isotope data ($^2$H and $^{18}$O) of the mobile flow domain (sampled with suction lysimeters) and the bulk soil water (slow plus fast flow domain) sampled with the direct equilibration method (Wassenaar et al., 2008) for benchmarking the model performance at the individual study sites as presented by Sprenger et al. (2018). Our previous study showed that the conceptualization of the subsurface with two pore domains that exchange water via the soil gas phase improved the simulation of the soil water stable isotopic composition at the investigated sites compared to an assumption of uniform flow. Therefore, we apply the same model set up of SWIS as presented in detail by Sprenger et al. (2018). The model domain covered the soil profile down to 50 cm depth in 5 cm intervals. Root water uptake was limited according to rooting depth observations to the upper 15 cm at the heather site in Bruntland Burn and to the entire 50 cm soil profile at the forested sites. Soil evaporation (E) was limited to the upper 10 cm based on experiments by Or et al. (2013). ET was partitioned into E and transpiration (T) according to the canopy coverage. Since sap flow was measured at the forest site in Bruntland Burn (Wang et al., 2017a) and E estimates based on the maximum entropy theory were available for the heather site in Bruntland Burn (Wang et al., 2017b), we used this information to adjust the partitioning of ET at these sites. E and T both decreased linearly over depth and occurred from both the fast and slow flow domains (T limited to the permanent wilting point assumed to be at 15000 hPa, Figure S 2). Contrary to the application of the SWIS model for stable isotope modelling, E did not alter the virtual tracer concentration (similar to T), but reduced the soil moisture at the depths of E losses and root water uptake, respectively. Precipitation was divided into interception and throughfall according to the canopy coverage, and the surplus infiltrated into the soil when the interception storage capacity was reached.

## 2.4 Estimation of travel times and water ages

We defined the start of travel times and zero water age of waters as the day of infiltration into the soil profile. To derive the travel times and water ages, we ran the model for each day of rainfall or snowmelt from 06/2011 for Bruntland Burn and Dorset and from 01/2010 for Krycklan to 09/2016 and tracked the fate of a virtual tracer in soil storage (fast flow domain, slow flow domain and total storage) and water fluxes (E, T and R) as suggested by Sprenger et al. (2016). The number of days with rainfall or snowmelt of all days of simulation were 1381/1943 for the Bruntland Burn sites, 684/1984 for Dorset and 801/2465 for the Krycklan site.

In line with the definitions by Benettin et al. (2015), we consider two different metrics as conceptualized in Figure 1. The first was the median travel time (MTT) as a forward approach that estimates how long it takes the infiltrated water take to leave the soil as E, T or R flux. The second was the median water age (A) as a backward approach estimating





the age of water in the output fluxes and the soil storage since it infiltrated into the soil. To derive the MTT, we extracted the breakthrough curves of each virtual tracer mass ($I_j$) introduced during individual infiltration events in fluxes of E, T and R at the bottom of the profile. Tracer concentrations $O_j(t)$ in the output fluxes for each day after introduction of the virtual tracer $I_j$ at time $t_0$ were normalized by the infiltrated tracer mass ($O_j(t)/I_j(t_0)$, Figure 1 left).

We then computed the median of the individual breakthrough curves, which then described the time it took until 50 % of the infiltrated water ended up in the considered output flux from the soil (Sprenger et al., 2016). This leads to a time series showing the median travel times (MTT in days) required to leave the system either via evaporation ($MTT_E$), transpiration ($MTT_T$), or recharge ($MTT_R$). Since MTT would be underestimated if the cumulative normalized breakthrough curve of the virtual tracers would not reach unity, we limited the MTT analysis to the period from 2012-

2015. We further used the individual breakthrough curves in the E, T and R fluxes to derive master travel time distributions (MTTD) as introduced by Heidbüchel et al. (2012). In line with Heidbüchel et al. (2012), we superimposed all individual breakthrough curves, weighted them by the event size and normalized them by the total introduced virtual tracer mass. Such a weighted average travel time distribution was derived for evaporation ($MTTD_E$), transpiration ($MTTD_T$), and recharge ($MTTD_R$) fluxes at each study site. The time after which 50 % of the average

tracer mass has left via the considered flux was defined as the median of the MTTD.

Water ages, in contrast, were derived from the relative tracer mass of individual infiltration events ($I_j$, $I_{j+1}$, $I_{j+2}$, $I_{j+3}$, $I_{j+4}$ in Figure 1) based on the normalized concentrations ($O_j(t)/I_j(t_0)$) in the soil storage and the output fluxes. The median water age was then defined as the 0.5 percentile of the cumulative mass weighted water age distribution in the considered storage or flux. This way, water of unknown age stored in the hydrologic system from the time before the

simulation period has little effect on the water age metric. However, to prevent bias due to such water, we limited the water age analysis to the period from 2013-2016. Here, we report the median water ages in the fast flow domain ($A_{Sf}$), the slow flow domain ($A_{Ss}$), and total soil storage ($A_{St}$) and of the evaporation ($A_E$), transpiration ($A_T$) and recharge fluxes ($A_R$).

Distributions of travel times and water ages in fluxes and storages were derived using cosine kernel density estimations

(Venables and Ripley, 2011). Differences in MTT and water ages between the four sites were analysed using the non-parametric Kruskal-Wallis test with a post-hoc Dunn test (significance level of 0.05), since the data were not normally distributed according to the Shapiro-Wilk test. Since running the model for the considered 5 years took between 1 and 3 h and our analysis required running the model between 684 and 1381 times (for each day of precipitation), we were not able to do a formal uncertainty analysis due to the long computation times (up to >100 d for one set of MTT and

water ages per site). We therefore limit the presented water age analysis to one realization using a parameter set that was previously shown to reflect the water flow and transport dynamics well using stable isotope data (Sprenger et al., 2018).





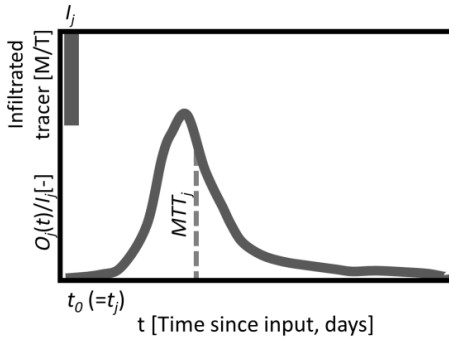
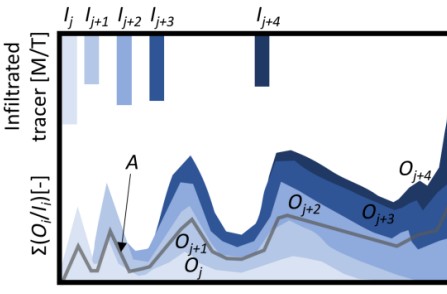

Figure 1 Conceptual visualization of the procedure used to derive median travel times (MTT) of output fluxes (i.e., evaporation, transpiration, recharge) (left plot) and median water ages (A) in the output fluxes or the soil storage (total storage, storage in fast and slow flow domain, respectively) (right plot). For MTT, the breakthrough of an infiltrated virtual tracer ($I_j$) introduced for each individual day of rainfall/snowmelt was traced in each output flux ($O_j(t)$). The median of the normalized outflow mass flux describes the time until half of the total tracer mass leaving the soil via the output flux was reached. The median water ages in fluxes and storage were derived from the age of the 0.5 percentile of the cumulative age distribution of individual tracer inputs (e.g., $I_j$ to $I_{j+4}$) in the considered flux or storage, visualized as the grey line.

## 3    Results

### 3.1    Travel times (How long does it take for infiltrated water to leave the soil again?)

#### 3.1.1    Travel times for evaporation flux

The median estimated travel time for infiltrated water until it was evaporated ($MTT_E$) varied usually between 4 and 13 days, but was occasionally older than 60 days during late autumn and winter at the Krycklan site (Figure 2a). In these cases, the fast flow domain dried out and relatively old water from the slow flow domain evaporated. $MTT_E$ tended to be greater when water infiltrated during periods of limited E and infiltration. $MTT_E$ values were similar across the sites (see distribution plots in Figure 2a), and there were no significant differences between the sites in Bruntland Burn and as well as between the forested site in Bruntland Burn and Dorset. However, at Krycklan, where travel times of > 100 days occurred when the fast flow domain run dry, the $MTT_E$ was significantly different to the other sites (Table 1).

#### 3.1.2    Travel times for transpiration flux

Median travel time of infiltrated water before it was taken up by the roots ($MTT_T$) was estimated to vary between a few days for waters infiltrated during the growing season and up to 250 days when the water infiltrated just after the growing season (Figure 2b). Thus, water introduced when the vegetation was active was quickly taken up by the plants leading to low $MTT_T$. However, when water infiltrated during the dormant season, this water aged in the rooting zone until it was transpired in the following spring. This resulted in a generally decreasing trend of $MTT_T$ towards the onset of the growing season. $MTT_T$ dynamics were similar across the four sites, due to similar seasonal climatic conditions and growing season length. However, shallower rooting depths for the heather site limited the water uptake to waters





of relatively shorter travel times as the shrubs did not have access to water in deeper soils with longer travel times (red triangles in Figure 2b). Therefore, $MTT_T$ at site the heather site in Bruntland Burn was significantly shorter than for the forested site there, which experienced the same climate forcing but had a rooting depth of 50 cm (Table 1). $MTT_T$ and $MTT_E$ did not show a correlation but had different dynamics, because the seasonal T flux was much larger than E during the growing season, while the E flux generally remained relatively small throughout the year (Figure S 4).

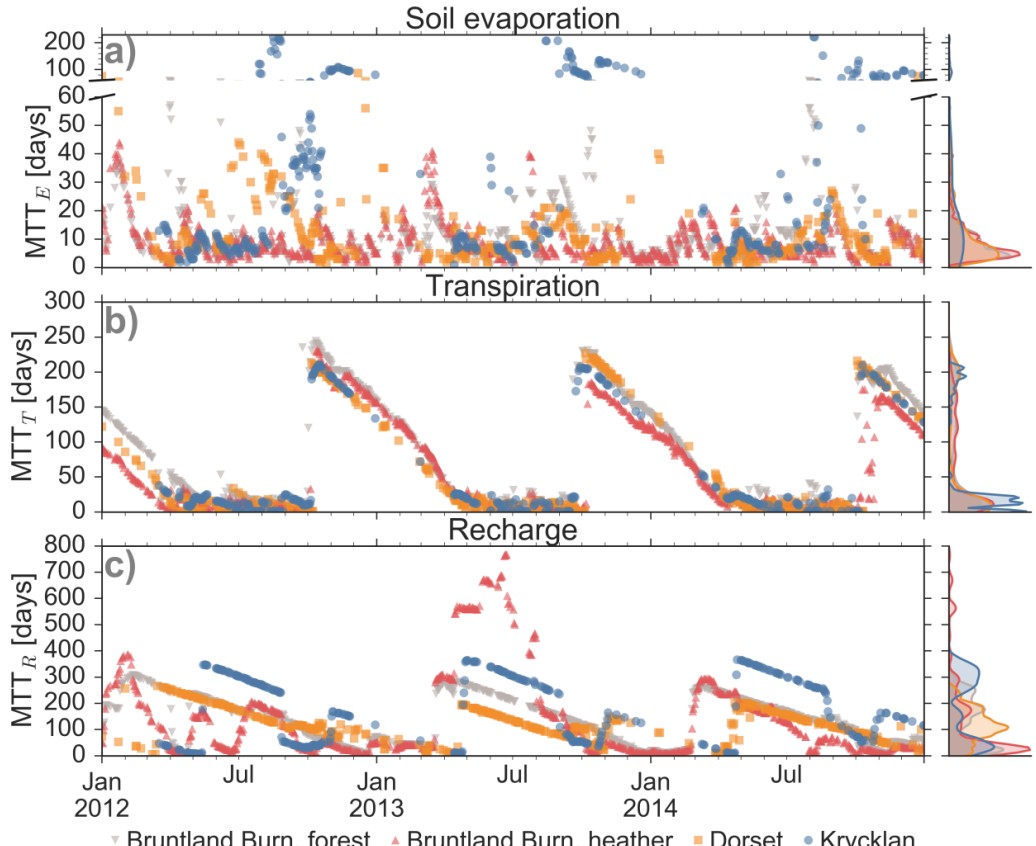

Figure 2 Median travel times (MTT) of water infiltrating into the soil on a specific day (given on the x-axis) until leaving the soil (a) via soil evaporation ($MTT_E$), (b) transpiration ($MTT_T$), or (c) recharge ($MTT_R$) flux. Colour code according to the four studied sites. Note that for days without precipitation or snowmelt, no travel times could be calculated. In subplot (a), y-axis has different scales for $MTT_E < 60$ days and >60 days. Density distributions of the travel times are shown for each site on the right-hand side.

### 3.1.3    Travel times for recharge flux

Median travel times for water to pass the 50 cm soil depth ($MTT_R$) showed a clear seasonal pattern with longest travel times (200 to 600 days) for water that infiltrated at the end of spring. In contrast, shortest $MTT_R$ (<50 days) prevailed during spring (Dorset and Krycklan) and winter (Bruntland Burn) (Figure 2c). Thus, water that infiltrated during periods of relatively low wetness in the snow-dominated sites in Dorset and Krycklan or rainfall that fell during the growing season when T rates were highest in Bruntland Burn had the longest R travel times (Figure S 5). While $MTT_R$



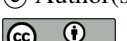

was not related to the R flux on the day of the traced infiltration event, $MTT_R$ was mainly governed by the time until major flushing of the soil water storage occurred (Figure 3): the longer it took to for intense R fluxes (defined as >1.5 mm day$^{-1}$) to occur following the traced water infiltrating into the soil, the longer it took for the water to become recharge. While the $MTT_R$ values were similar for the forested site in Bruntland Burn and Dorset, $MTT_R$ at the heather

site in Brunland Burn was significantly shorter and at Krycklan significantly longer than at the forested Bruntland Burn and Dorset sites (Table 1).

Table 1 Summary of median travel time (MTT, shown in Figure 2) and master travel time distribution (MTTD) characteristics of the four study sites: median of MTT (25$^{th}$ percentile, 75$^{th}$ percentile) in the evaporation flux ($MTT_E$), transpiration flux ($MTT_T$), recharge flux ($MTT_R$), median of the MTTD of the evaporation flux ($MTTD_E$),
transpiration flux ($MTTD_T$), recharge flux ($MTTD_R$). Letters as superscript indicate significant differences in each column. Sites with the same letter are not significantly different regarding the MTT or MTTD of the considered flux.

| Site | $MTT_E$ [days] | $MTT_T$ [days] | $MTT_R$ [days] | $MTTD_E$ [days] | $MTTD_T$ [days] | $MTTD_R$ [days] |
|---|---|---|---|---|---|---|
| Bruntland Burn, forested | 8 (5, 13)[AB] | 44 (13, 149)[A] | 131 (41, 203)[A] | 7[A] | 27[A] | 50[A] |
| Bruntland Burn, heather | 7 (5, 11)[A] | 27 (10, 123)[B] | 51 (24, 183)[B] | 8[A] | 17[B] | 21[B] |
| Dorset | 8 (4, 14)[B] | 13 (1, 130)[C] | 112 (79, 172)[A] | 6[A] | 19[A] | 77[C] |
| Krycklan | 13 (7, 75)[C] | 18 (10, 31)[C] | 158 (42, 298)[C] | 17[B] | 17[C] | 34[A] |

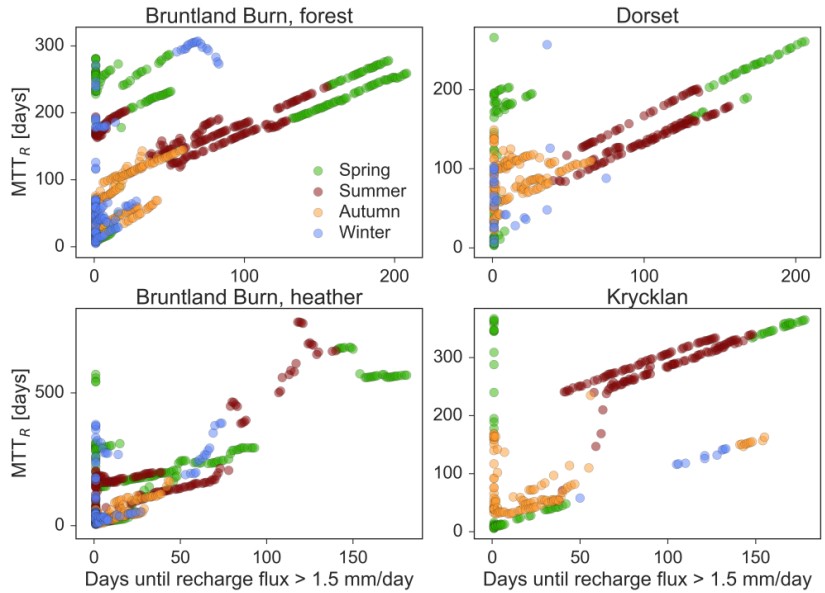

Figure 3 Median travel times in recharge flux ($MTT_R$) for each study site as a function of days required to produce
intense recharge fluxes (here defined as >1.5 mm day$^{-1}$) following infiltration of the traced water into the soil. Colour code represents the season when the traced water infiltrated the soil.





### 3.1.4    Master travel time distributions

The weighted average description of the travel time, as the master travel time distribution (MTTD), showed the general differences between water transport via E, T and R (Figure 4). Fastest transport of infiltrated was generally for the E flux with $MTTD_E$ showing response within one day and relatively short tails of the distribution (dashed lines in Figure 4). $MTTD_T$ also showed relatively quick response, but the decrease in tracer mass over time was lower than for $MTTD_E$. The time until the virtual tracer was observed in the R flux ($MTTD_R$) was generally longer compared to the fluxes to the atmosphere and the distributions were characterized by long tails. $MTTD_E$ at Krycklan was significantly different from the other sites with a median of 17 days compared to 6-8 days at Bruntland Burn and Dorset (Table 1). $MTTD_T$ were statistically similar for the forested site in Bruntland Burn and Dorset, but significantly different to the $MTTD_T$ at the heather site in Bruntland Burn and Krycklan site. $MTTD_R$ at the heather site was also significantly different and its median with 21 days was shortest compared to the other sites. At Dorset, median $MTTD_R$ were longest (77 days) and their distribution significantly different from the other sites, while $MTTD_R$ for the forested Bruntland Burn and Krycklan sites were statistically similar with median values of 50 and 34 days, respectively (Table 1).

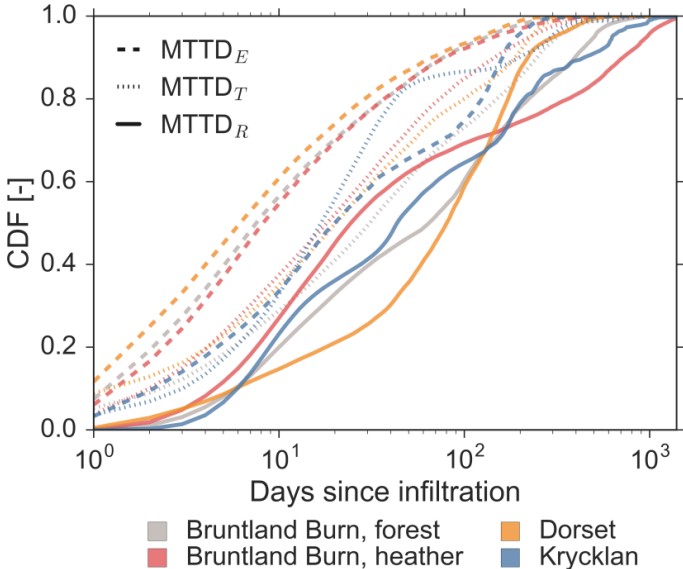

Figure 4 Cumulative density function of the master transit time distributions (MTTD) of the evaporation flux (dashed lines, $MTTD_E$), transpiration flux (dotted lines, $MTTD_T$), recharge flux (unbroken lines, $MTTD_R$) for the four study sites (color code).

### 3.2    Water ages (What are the ages of the storage in the soil and in the fluxes leaving the soil?)

### 3.2.1    Water ages of soil storage

The median age of the total water stored in the simulated 50 cm soil profile ($A_{St}$) ranged from few days to 300 days (Figure 5a). Short-term dynamics of $A_{St}$ were driven by the infiltration patterns with generally smaller $A_{St}$ after high infiltration rates (Figure S 6). $A_{St}$ generally increased during periods of low infiltration such as dry summers at Bruntland Burn and Krycklan or throughout snow cover at Dorset and Krycklan. $A_{St}$ was usually larger for lower storage volumes and decreases exponentially with increase in soil storage. This inverse storage relationship was most





pronounced for the water ages in the fast flow domain ($A_{Sf}$ in Figure 6). Exceptions of low $A_{Sf}$ during low storage occurred when the fast flow domain dried out and was then refilled by young waters during infiltration events (see several red and orange data points in the first row in Figure 6). $A_{Sf}$ was generally smaller than $A_{Ss}$ (Figure 5c and d). Dynamics of $A_{Sf}$ were generally highly responsive to infiltration, but the response of $A_{Ss}$ was usually less and often

delayed compared to $A_{Sf}$. More intense short-term dynamics in $A_{Ss}$ – and consequently also in $A_{St}$ – were limited to sites and periods when the fast flow domain was empty (e.g., July 2013 and 2014 at the forested site in Bruntland Burn and summers at Dorset, Figure 5). $A_{Ss}$ was generally larger and more damped than $A_{St}$ (Figure 5).

$A_{St}$ was significantly different between all sites (Table 2). For $A_{Ss}$, the Bruntland Burn sites did not differ significantly, probably due to the same climatic forcing and similar shape of the water retention curve for the slow flow domain in

the upper horizon (Figure S 2). For $A_{Sf}$, the forested site in Bruntland Burn and in Dorset were not significantly different, as their water retention for the fast flow domain was similar with drying out of the fast flow domain during summer. This decrease in the water storage of the fast flow domain led to large $A_{Sf}$ just before running dry due to interaction with older water in the slow flow domain.

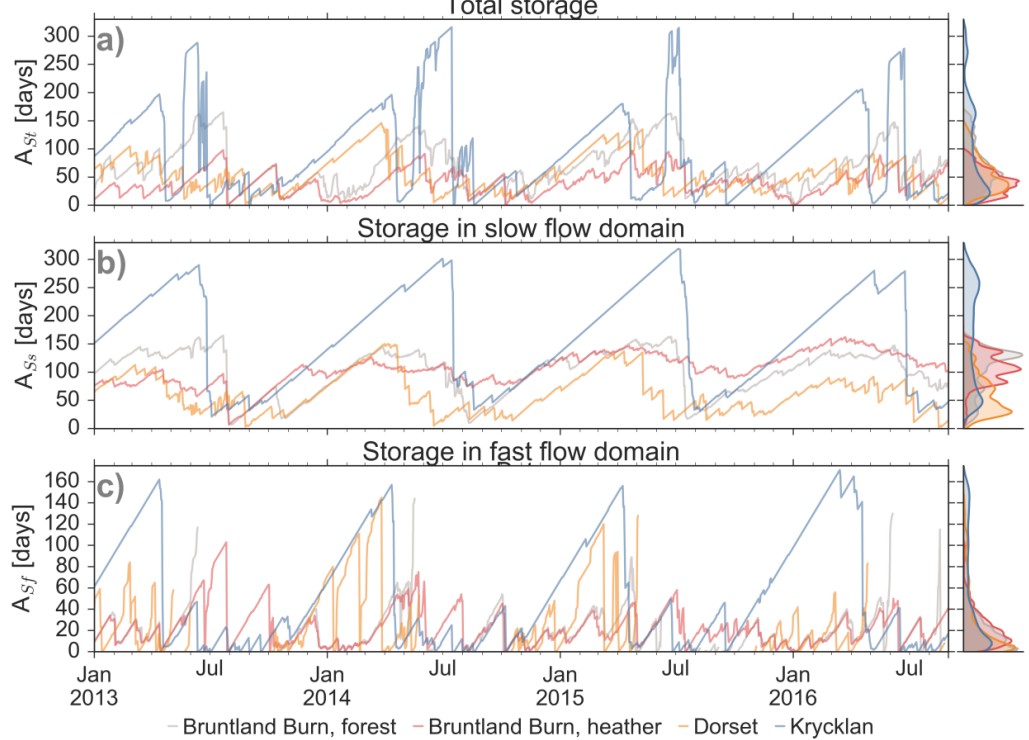

Figure 5 Water ages of (a) total soil water storage ($A_{St}$), (b) storage in the slow flow domain ($A_{Ss}$), and (c) storage in the fast flow domain ($A_{Sf}$). Note that when storage in the fast flow domain is zero, there is no water age for the storage. For site NH, the spin up period of 1.5 years was not sufficient to replace the water in the slow flow domain, resulting in continuously increasing water ages for $A_{St}$ and $A_{Ss}$, which inhibits their analysis (dashed lines). The colour code is according to the four study sites. Density distributions of the water ages are shown for each site are shown on the
right-hand side.



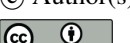

$A_{St}$ was usually <120 days in the upper 5 cm and generally increased linearly with depth over the rooting zone (Figure 7). The greatest variability in $A_{St}$ was at the forested sites at depths <25 cm, where site-specific $A_{St}$ maxima occurred during the growing season while $A_{St}$ < 60 days happened during periods of high recharge in the dormant season and during snowmelt when soil waters were well connected throughout the profile. At the heather site in Bruntland Burn

maximum $A_{St}$ was found just below the rooting zone (15 cm) and not at the bottom of the soil profile (Figure 7). Thus, soil water storage volumes, altered by the root water uptake, affected the water transport and mixing processes such that younger water in the fast flow domain by-passed the older water stored in the slow flow domain.

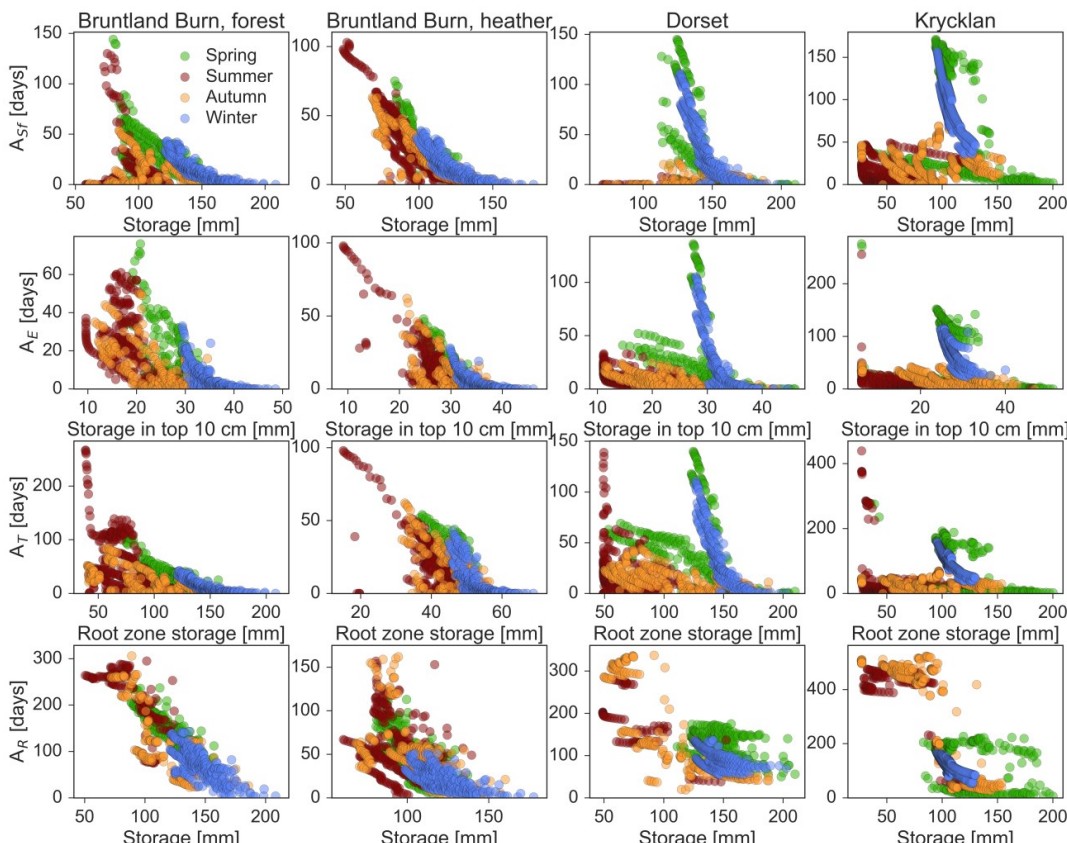

Figure 6 Median water age of fast flow domain ($A_{Sf}$, first row), evaporation ($A_E$, second row), transpiration ($A_T$, 3rd
row), and recharge flux ($A_R$, last row) as a function of the water stored in the entire soil, in the upper 10 cm, in the root zone and in the entire soil profile, respectively. Each column represents one of the four studied sites and the colour code represents the season.

### 3.2.2 Water ages of evaporation flux

The median age of the water in the E flux ($A_E$) ranged between 0 days and 140 days with largest values during periods
of snow cover at Dorset and Krycklan (Figure 8a). $A_E$ was exponentially related to storage in the upper 10 cm from which E occurred (Figure 6). For Krycklan and Dorset, this exponential relationship was most pronounced for periods of decreasing storage during snow accumulation in winter, when oldest E fluxes were observed. $A_E$ was largest for periods of minimal infiltration and decreased exponentially with increasing infiltration rates (Figure S 7). Due to the





same climatic conditions at the heather and forested sites in Bruntland Burn, $A_E$ values were not significantly different and on average lower than at the Dorset and Krycklan sites (Table 2).

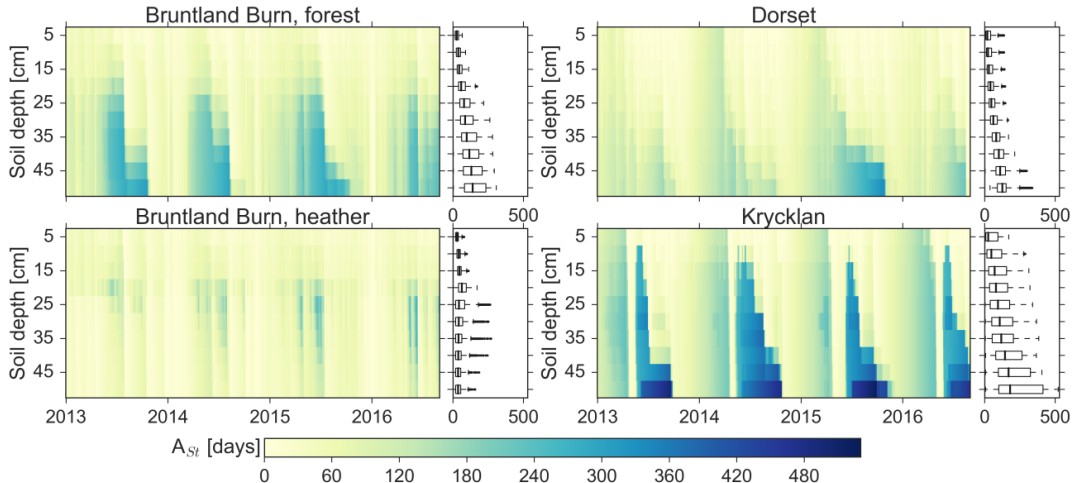

Figure 7 Median water age of the total soil water ($A_{St}$) at 5 cm intervals over the simulation period for each of the study sites.

### 3.2.3 Water ages of transpiration flux

Water ages in T ($A_T$) ranged between 0 days and up to 300 days and showed similar dynamics to those of $A_E$ for most periods (Figure 8b). However, $A_T$ was usually older than $A_E$, since the root water uptake included deeper soil layers than E did. $A_T$ decreased with increasing storage volume (Figure 6), $A_T$ declined after precipitation events added young water to the rooting zone during summer and early autumn when storage was generally low. The inverse storage effect between root zone storage and $A_T$ was most pronounced at Bruntland Burn, where infiltration regularly occurred throughout the year. Under these conditions, the largest $A_T$ occurred when the soil dried out and root water uptake of deeper soil layers became more relevant, leading to an increased relative contribution of older waters to modelled T. For Dorset and Krycklan, the oldest $A_T$ values were not only related to low root zone storage, but also to the aging of water in the root zone during snow cover in winter when infiltration rates were low and transpired water thus increasingly became older with time. $A_T$ had a linear relationship with $A_{Sf}$ during wet periods, but approached $A_{Ss}$ during dry periods, due to a shift in root water uptake from the fast flow domain to uptake largely from the slow flow domain. $A_T$ was significantly smaller at the heather site in Bruntland Burn, where root water uptake was limited to the upper 15 cm, than compared to the forested sites with rooting depths down to 50 cm (Table 2). $A_T$ did not differ significantly between the forested site in Bruntland Burn and Dorset, but was significantly higher at Krycklan, where the oldest water was stored in the soil.

### 3.2.4 Water ages of recharge flux

Median water age of the R flux through the 50 cm depth plane ($A_R$) were generally exceeded, but was usually linearly related to, the total soil storage water age ($A_{St}$) (cf. Figure 8 and Figure 4). However, this linear relationship did not hold for periods of low R flux, and $A_R$ became $\gg A_{St}$. (e.g. summer for Krycklan in Figure 8c). $A_R$ had a strong relationship with R flux and total water storage: the oldest water was recharged during low R fluxes and low storage




volumes, respectively (Figure 6). As $A_R$ was generally strongly related to the age dynamics of the fast flow domain, the differences in $A_R$ among the sites were similar to the differences in $A_{Sf}$, with $A_R$ being significantly lower at the heather site in Bruntland Burn and significantly higher at the Krycklan compared to the forested site in Bruntland Burn and Dorset (Table 2).

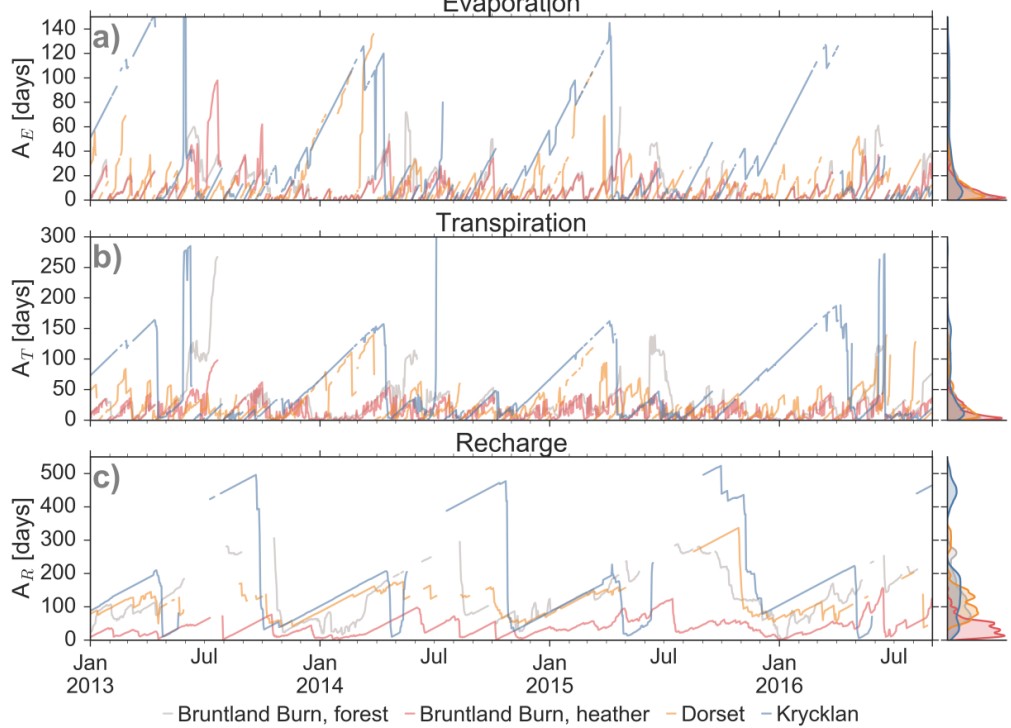

Figure 8 Median water ages of (a) evaporation ($A_E$), (b) transpiration ($A_T$), and (c) recharge flux ($A_R$). Note, that when flux is zero, there is no water age for the flux. The colour code is according to the four study sites. Density distributions of the water ages are shown for each site are shown on the right-hand side.

Table 2 Summary of water age characteristics of the four study sites: median (25th percentile, 75th percentile) of the median water ages in the total storage ($A_{St}$), storage in slow flow domain ($A_{Ss}$), storage in fast flow domain ($A_{Sf}$), evaporation flux ($A_E$), transpiration flux ($A_T$), and recharge flux ($A_R$). Letters as superscript indicate significant differences in each column. Sites with the same letter are not significantly different regarding the water ages of the considered storage or flux.

| Site | $A_{St}$ [days] | $A_{Ss}$ [days] | $A_{Sf}$ [days] | $A_E$ [days] | $A_T$ [days] | $A_R$ [days] |
|---|---|---|---|---|---|---|
| Bruntland Burn, forested | 61 (39, 94)[A] | 114 (70, 132)[A] | 15 (6, 27)[A] | 8 (3, 19)[A] | 19 (8, 34)[A] | 110 (70, 162)[A] |
| Bruntland Burn, heather | 39 (24, 55)[B] | 107 (87, 126)[A] | 18 (10, 31)[B] | 7 (3, 13)[A] | 11 (4, 25)[B] | 36 (19, 54)[B] |
| Dorset | 48 (31, 74)[C] | 56 (32, 85)[B] | 14 (4, 42)[A] | 10 (4, 19)[B] | 18 (8, 38)[A] | 106 (79, 143)[A] |
| Krycklan | 76 (31, 155)[D] | 164 (76, 242)[C] | 31 (9, 94)[C] | 22 (7, 63)[C] | 31 (11, 96)[C] | 150 (85, 220)[C] |




## 4    Discussion

Our simulations emphasize the time variant character of water ages and travel times in hydrological systems, as observed at the catchment scale in various recent studies (introduced in section 1). Further, the age dynamics in the soil waters were driven by the variability in water stored in the soils, supporting an "inverse storage effect" as discussed

by Harman (2015). Numerical modelling using SWIS provided us with new insights into how different pore spaces, respectively representing fast and slow flow domains, differ from each other in terms of water ages. We further showed how the age variability over the soil pores affects the water ages of the associated fluxes. Since all travel times and water ages depend on the water stored in the soil, we will first discuss this and then include E, T and R travel times and water ages.

### 4.1    What controls soil water storage and water ages?

As the age of the total soil water as well those of the fast and slow flow domains ($A_{St}$, $A_{Sf}$, $A_{Ss}$, respectively) generally decreased with increasing storage volume, the antecedent hydro-meteorological conditions controlled the soil water age dynamics. For periods with high ET fluxes, that led to low storage volumes, water ages in the soil pores generally increased over time (Bruntland Burn sites in Figure 6), because the youngest water left the soil column preferentially

via ET. Thus, E rates and vegetation uptake directly impacted water age dynamics in the critical zone. However, periods of low ET could also lead to higher soil water ages, as when winter snow accumulation resulted in the cessation of infiltration and R fluxes (Dorset and Krycklan sites in Figure 6). Now, water resided in the soil and aged over time towards spring (Figure 5). Once snowmelt began, soil water storage increased and infiltrating melt water lead to the lowest soil water ages at Dorset and Krycklan. Both the ET-driven and the snowmelt-driven cases, result in an inverse

storage effect, where water in the soil became younger for higher soil water volumes. For water flow in the vadose zone, as conceptualized according to van Genuchten (1980) in the SWIS model, higher soil moisture generally leads to higher water mobility, as the hydraulic conductivity decreases sharply when the soil dries out (decreasing pressure heads). Thus, water that infiltrates during high storage volumes is more likely to be transported more rapidly than water that infiltrates during periods of low storage. As a result, young water ages prevailed across the entire 50 cm

soil profile during periods of high storage (Figure 7). Note that this conceptualization would not hold when soil dryness induces preferential flow due to water repellency (hydrophobicity) (Ritsema et al., 1993; Weiler and Naef, 2003).

The impact of increased water mobility (i.e., flushing) on soil water ages during greater soil wetness is supported by stable isotope data ($^2$H and $^{18}$O). For the Bruntland Burn sites, Sprenger et al. (2017) showed that the isotopic variability in bulk soil water was greatest after intense infiltration events, revealing that event water mixed effectively

with pre-event water in the upper 20 cm. Also isotope data from mobile soil water (Peralta-Tapia et al., 2015) and bulk soil water (Sprenger et al., in review) at the Krycklan site had a strong relationship with the isotopic compositions of previously infiltrated water, which shows that a high proportion of the soil (pre-event) water is replaced by or mixes with recent event water. Such isotope studies provide a snapshot view of water transport in the field, but modelling approaches benchmarked against or calibrated on such observations, like our study, allow insights into short term

dynamics. Our modelling approach also enabled us to assess potential differences in water ages within the pore space due to the conceptualization of a fast and slow flow domain.



While the inverse storage effect occurred for both, the fast and the slow flow domain, the time scales and dynamics were different with smaller and more responsive $A_{Sf}$ relative to $A_{Ss}$. (Figure 5). Water fluxes were generally slower in the slow flow domain, due to higher pressure heads (h>600 hPa, Figure S 2) and their control on the hydraulic conductivity (van Genuchten, 1980). Thus, $A_{Ss}$ (and the storage in the slow flow domain) does generally not change

as rapidly as $A_{Sf}$, which is influenced by highly variable storage volumes in the fast flow domain. However, since the ratio between water stored in the fast and slow flow domain changes as a function of soil wetness (Sprenger et al., 2018), the impact of the two domains on total soil water age $A_{St}$ also varies over time. During summer, when the storage in the fast flow domain decreased or was even fully depleted (at forested Bruntland Burn and Dorset sites), $A_{St}$ approached or equalled $A_{Ss}$. Due to exchange between fast and slow flow domain, $A_{Sf}$ approached $A_{Ss}$ just before

the fast flow domain dried out (see forested Bruntland Burn and Dorset in Figure 5c). Our age analyses therefore support the hypothesis by Sprenger et al. (2017) that old water in smaller soil pores can lead to a "memory effect" in the bulk soil water isotope compositions. Such a "memory effect" was further shown to lead to a lagged response of the soil water stable isotope compositions to hydro-meteorological forcing at five long-term experimental catchments in northern environments (Sprenger et al., in review). Further, observed differences in the isotopic compositions of

mobile and bulk soil water in the field were often related to the potential age differences of waters sampled at different mobilities (Landon et al., 1999; Brooks et al., 2010; Geris et al., 2015; Sprenger et al., 2015a; Oerter and Bowen, 2017). Our results and recent simulations by Smith et al. (2018) support such interpretations, as the water in the slow flow domain was generally older than the water in the fast flow domain ($A_{Ss} > A_{Sf}$). However, since the differences between $A_{Ss}$ and $A_{Sf}$ were variable in time and were often maximized in early spring, such anomalies in water ages

are likely to be reflected in the isotopic compositions of the water, with the older water in small pores being less depleted in heavy isotopes (originating partly from autumn precipitation) than the young water in larger soil pores draining recently infiltrated isotopically depleted snowmelt or winter precipitation. Such isotopic differences resulting from different water ages affect our interpretation of soil water stable isotopes sampled either with suction lysimeter (mobile water in the fast flow domain) or cryogenic vacuum extraction (bulk soil water in fast and slow flow domain).

For example, Brooks et al. (2010) reported different isotopic compositions for mobile and bulk soil water samples, which led to the formulation of the two water world hypothesis (TWW) (McDonnell, 2014). In a TWW scenario, tightly bound soil water is not displaced via translatory flow, does not mix with or displace mobile water, and does not enter the stream. However, experimental work recently showed that there is interaction between mobile and less mobile soil waters (Vargas et al., 2017), as conceptualized in the applied SWIS model. Our simulations further

question if water in the slow flow domain - as defined in SWIS - will not eventually recharge the groundwater and streams. The virtually introduced tracer eventually disappears from the soil water storage of the slow flow domain, due to loss of the tracer to the atmosphere (ET flux), interaction with the fast flow domain and recharge within the slow flow domain. Nevertheless, while definition of the slow flow domain using a higher threshold pressure head (e.g. field capacity as suggested by Brantley et al. (2017) rather than the currently assumed 600 hPa) would result in its

water becoming more tightly bound, interaction with more mobile waters would likely still persist (Vargas et al., 2017).





## 4.2     What controls travel times and water ages in evapotranspiration?

Since water loss from soil storage as E flux was limited to the upper 10 cm in our simulations, travel times and water ages are directly related to the water age dynamics in the top soil. In contrast, the rooting zone covered the entire soil profile for the forested sites and down to 15 cm soil depth for the heather site, which affected the resulting travel times

and water ages of T accordingly.

While investigation of water ages in the ET flux is relatively new (Botter et al., 2011), age dynamics have usually been assessed for the bulk ET flux (Harman, 2015; Soulsby et al., 2015; van der Velde et al., 2015; van Huijgevoort et al., 2016; Soulsby et al., 2016). While it was shown that tracer-aided modelling using stable isotopes of water benefits from partitioning ET into a fractionating E flux and a non-fractionating T flux (Knighton et al., 2017), separate

water age analyses for E and T have been considered only recently (Smith et al., 2018). However, our analyses showed that the two different fluxes can have markedly different travel time dynamics (Figure 2), average travel time distributions (Figure 4), and water age dynamics (Figure 8). Thus, our process understanding of how vegetation affects water ages in hydrological systems would particularly benefit from further assessments of the differences between E and T water ages. Such investigations are of special interest in light of ongoing research regarding the consequences

of a potential TWW hypothesis on water age estimations based on tracer-aided modelling (Hrachowitz et al., 2016). In particular, our simulations underline that ET fluxes do usually not withdraw water from a well-mixed pool, which is increasingly acknowledged in water age studies (Harman, 2015; Smith et al., 2018).

Similar to findings by Smith et al. (2018) for the heather site in Bruntland Burn, our estimates for $A_E$ at that site were highest during periods of limited infiltration (e.g. 10-year return period drought in summer 2013 at Bruntland Burn in

Figure 8a). However, E water ages reported by Smith et al. (2018) were higher than our estimates, which is probably due to their conceptualization of the subsurface into one domain with and one without downward flux, which resulted in generally higher water ages in the shallow soils compared to our estimates. Water age estimates by Queloz et al. (2015) for the ET flux from a lysimeter were less variable, but with about 10 to 20 days of the magnitude of our $A_E$ and $A_T$ estimates. The forward travel time distributions for water leaving the soil via ET presented by Queloz et al.

(2015) also showed shapes similar to our reported $MTTD_E$ and $MTTD_T$ with peaks in the first few days and tails of the distribution that can reach up to 200 days (Figure 4). We attribute the long tails of the $MTTD_E$ and $MTTD_T$ to both the ET flux from the slow flow domain and root water uptake from deeper soil layers.

With regard to $A_T$, our soil physical model showed a similar inverse storage effect as the approach using storage selection functions (Smith et al., 2018): water taken up by plants was generally younger during higher soil storage.

While Smith et al. (2018) had a dynamic root water uptake depth, T loss in the SWIS model decreases linear with depth as long as the pressure head does not reach the permanent wilting point, which is usually not reached at the investigated sites (Sprenger et al., 2018). Thus, it is likely that the differences between the dynamic root water uptake depths in the storage selection functions and the defined uptake profile in SWIS will be more pronounced when vegetation responds to intense drought by shifting the root water uptake to deeper soil layers (Volkmann et al., 2016).

A relationship between $MTT_T$ and T dynamics, with the onset and cessation of T at the beginning and the end of the growing season, has been shown previously (Sprenger et al., 2016), nevertheless, our experimental set up with two different vegetation types on similar soil types under the same climatic forcing in the Bruntland Burn reveals the




impact of rooting depth on the travel time dynamics. Median and maximum $MTT_T$ were shorter for the heather site than for the forested site in Bruntland Burn and the $MTTD_T$ had substantially different shapes at both sites with a lower median for the heather T travel times compared to the forest (Table 1). Our simulations generally showed that the ET water ages are not only affected by the soil water ages, but that vegetation and atmospheric demand in turn impact soil water age and travel time dynamics in a reciprocal manner.

### 4.3    What controls recharge travel times and water ages?

Water age and travel time dynamics of the recharge flux are the result of the interplay between the aforementioned linkages between soil water storage age and ET age dynamics. Since the estimated $MTT_R$ are a function of the subsequent recharge flux intensities (Figure 3), the probability of an introduced water parcel leaving the soil profile via recharge is higher during high flows. Such a relationship between forward travel times and the recharge flux dynamics were also found in modelling studies on a controlled lysimeter (Queloz et al., 2015) and 35 field sites in Luxembourg (Sprenger et al., 2016). While catchment scale travel time studies based on conceptual lumped models also showed that the subsequent precipitation patterns affect the travel time dynamics of runoff (Heidbüchel et al., 2013; Hrachowitz et al., 2013; Harman and Kim, 2014; Peters et al., 2014; Peralta-Tapia et al., 2016), our application of a 1D soil physical model provided insights into the processes in the upper critical zone leading to such behaviour at the plot scale. The simulations with SWIS highlight the effect of ET fluxes on recharge travel times, as both storage and recharge are influenced by ET rates. Consequently, one can see the exceptionally high $MTT_R$ for the few infiltration events during a 10-year return period dry episode in summer 2013 for the heather site at Bruntland Burn. Further, the seasonal decrease of $MTT_R$ due the preferential recharge during the dormant season (Bruntland Burn) or snowmelt (Dorset and Krycklan) emphasises the impact of ET on vadose zone travel times. Such a clear influence of vegetation on travel times in the catchment runoff is commonly not seen as the stream integrates water moving via different pathways and thus obscures any ET signal (Tetzlaff et al., 2014; Kirchner, 2016).

The conceptualization of fast and slow flow domains resulted in $MTTD_R$ that indicated a maximum probability of infiltrating water recharging from the soil within 3 to 10 days after infiltration, although the tails of the $MTTD_R$ revealed that replacement of all water (turnover time) can take up to 1000 days (Figure 4). Thus, our modelling approach of a two-pore domain enabled the representation of the short-term responses and the long-term memory of the recharge composition in a soil column. As a result, water ages ($A_{St}$) did not always increase with depth, but instead became almost uniform throughout the soil column during intense infiltration periods. Occasionally $A_{St}$ was smaller at the bottom of the profile relative to just below the rooting zone, mainly due to root water uptake dynamics at the heather site in Bruntland Burn (Figure 7). While $A_R$ was generally higher than $A_{St}$, consistent with Queloz et al. (2015), our soil physical modelling approach revealed how the water ages develop with depth and lead to the resulting $A_R$ dynamics.

The pronounced longer water ages of the slow compared to the fast flow domain are of great relevance for the interpretation of studies on travel times in vadose zone water fluxes. These investigations are often based on models calibrated with isotope data from samples taken with zero-tension lysimeters (e.g., Asano et al., 2002), wick samplers (e.g., Timbe et al., 2014), suction lysimeters (e.g., Muñoz-Villers and McDonnell, 2012; Tetzlaff et al., 2014; Hu et



al., 2015), or from the outflow of lysimeters (e.g., Stumpp et al., 2009; Stumpp et al., 2012). Such methods limit isotope sampling to the most mobile water in the soil (Sprenger et al., 2015a), which is represented as the fast flow domain in the current application of the SWIS model. According to our simulations, travel time studies based on the most mobile waters in the soil are likely to underestimate travel times and water ages in the recharge fluxes.

Consequently, the turnover time of the soil pores will be underestimated in such studies, which can then lead to the assumption that nutrients or contaminants located in the vadose zone will be flushed out more rapidly than they actually are.

The R water ages at 50 cm depth in this study are obviously younger than catchment scale runoff water ages (Soulsby et al., 2015; Ala-aho et al., 2017b; Benettin et al., 2017; Kuppel et al., 2018). Nevertheless, there are similarities

regarding the age dynamics. The plot scale R water ages based on SWIS show a similar pattern to catchment runoff water ages with generally increasing values throughout spring towards summer (decreasing storage) and lowest water ages during winter (highest storage) (Soulsby et al., 2015; Ala-aho et al., 2017b; Benettin et al., 2017). Catchment runoff ages and plot scale $A_R$ show also similar dynamics at Dorset, as Piovano et al. (in review) also reported generally increasing water ages in the runoff during the snow dominated winter, while runoff ages during summer were more

dependent on the rainfall pattern. We see also similarities at Krycklan with increasing water ages for catchment runoff (Ala-aho et al., 2017b) and soil water recharge throughout the snow dominated winter. However, while snowmelt leads to a rapid decrease in water ages at both the catchment and the plot scales, the cessation of the soil recharge flux in the summer in the 1D-simulations results in the largest $A_R$ during fall when older soil water becomes remobilized. In contrast, Ala-aho et al. (2017b) reported generally small runoff water ages during summer at Krycklan with few

peaks due to dry periods, probably due to young water contributions from small mires (Peralta-Tapia et al., 2015).

In general, the long tails of the $MTTD_R$ indicate that the soil storage can probably add to the commonly observed long tails of catchment scale travel time distributions (e.g., Godsey et al., 2010; Hrachowitz et al., 2010). Our plot scale soil hydraulic simulations can help to better understand the processes taking place within the catchment and constraining or benchmarking spatially distributed hydrological models (van Huijgevoort et al., 2016; Ala-aho et al.,

2017b; Kuppel et al., 2018). Such catchment models cannot account for the soil physical processes in a similar detail as a 1D-model due to computational limitations. However, our results imply that it might be worth adding a dual-porosity representation, similar to the conceptualization in SWIS, to the recently published EcH2O-iso (Kuppel et al., 2018).

### 4.4 Limitations and outlook

While we cannot provide uncertainty estimates for the presented travel times and water ages due to restrictions imposed by computation time, comparison with soil moisture and stable isotope data at each site (Sprenger et al., 2018) indicates that the SWIS model captures the water flow and transport processes well. However, model calibration using soil moisture and stable isotope data as suggested by Sprenger et al. (2015b), would supply the basis of an assessment of how different parameter sets impact the model performance and water age estimates. Such an approach

would provide site specific characterization of the soil physical properties and would likely improve simulations compared to the currently applied pedotransfer functions and measurements on soil cores.



Since the investigated northern environments seldom experience severe drought, plant growth is usually not water limited, as for example was shown by Wang et al. (2017a) for Bruntland Burn. Thus, the assumption of a linear decrease in root water uptake with depth appears to be reasonable for the current study sites. However, several isotope studies have shown that, the root water uptake profile does not coincide with the root distribution if plants experience

water stress (e.g., Kulmatiski and Beard, 2013; Ellsworth and Sternberg, 2015; Volkmann et al., 2016). Hydraulic lift can further increase the complexity of soil-plant interaction, as experimentally observed and implemented in a soil hydraulic model by Meunier et al. (2018). Thus, an improved representation of such dynamics in the water uptake depths would be beneficial for modelling studies in arid environments. For the northern environments considered here, variability in the depths from where the plants take up their water appears to be limited (Smith et al., 2018).

The presented simulations are further limited by the discretization of the soil profile into 5 cm intervals, as this might be too coarse for an adequate representation of the interactions between atmospheric demand, T losses and mixing down the profile. However, computational limitations did not allow a smaller discretization and our study aimed to test the assumption that soil storage is a well-mixed water source of ET fluxes.

Lastly, while the investigated sites are not located on steep slopes, the 1D simulation cannot account for lateral flows

in the vadose zone that may potentially occur during extreme rain events (Soulsby et al., 2017). We further have to assume that all R flux leaving the soil profile will end up in groundwater, but spatially distributed catchment models (Ala-aho et al., 2017b; Kuppel et al., 2018) might reveal that such water could end up in the ET flux from saturated areas in the valley bottom when the groundwater feeds the riparian zone.

## 5   Conclusion

We have provided unique insights into the water ages of the upper critical zone using the soil physical model SWIS by tracking water through the soil profile and its associated fluxes from the soil at four investigated sites. Based on these 1D simulations, we revealed that the recently described inverse storage effect for catchment and hillslope runoff not only holds for recharge from the soil, but is also present for the transpiration and evaporation fluxes: water leaving the soil via evaporation, transpiration or recharge was generally younger the greater the soil water storage. The

conceptualization of the vadose zone into slow and fast flow domains and its discretization over depth allowed us to investigate how the water ages in the hydrologic fluxes develop over time and space based on the soil water volume and its age. The temporal age dynamics are mainly related to soil water storage dynamics. Thus, the seasonality of evaporation and vegetation activity according to the growing season affected the water ages in the soil. Future climate warming or vegetation cover change in the northern latitudes will thus directly affect critical zone water age dynamics.

Contrary to the common approach of employing bulk ET in water age analysis, we demonstrated that evaporation and transpiration have different water ages and travel times. Thus, an improved partitioning of the two fluxes appears to be essential to understanding the differential impact of evaporation, usually of relatively young waters from the top of the unsaturated zone, and transpiration, which can access older water from deeper soil layers, on water age dynamics. Furthermore, rooting depth was found to affect the transpiration water ages and travel times, with younger water ages

and shorter travel times for the shallow roots of heather relative to deeper-rooting trees. While both evaporation and transpiration generally have relatively young water ages and short travel times, the travel time distributions revealed



that the ET flux also contains considerably older (> 100 days) waters. We relate these old waters to the conceptualization of the subsurface into two pore domains. Water in the fast flow domain was usually about half as old as in the slow flow domain, which was fully exchanged within 1000 days and thus was not immobile. Nevertheless, the differences between the slow and fast flow domain are crucial for the interpretation of previous travel time studies

that have based their calibration on tracer data from the fast flow domain (e.g., suction lysimeter samples), since such studies will have underestimated travel times and water ages. Recharge travel times were mainly governed by the subsequent recharge flux dynamics in our study, and decreased during periods of intense flushing of the soil water during winter in Bruntland Burn and snow melt in Krycklan and Dorset. Transpiration travel times were controlled by vegetation phenology and the associated annual climatic cycle, with longest travel times for waters infiltrated at the

beginning of dormancy and short travel times throughout the growing season.

Our simulations generally extended insights on the water flow and transport processes from snap shot isotope sampling to new insights into both the seasonal and short-term dynamics of water ages in the critical zone. The soil physical simulations showed that the inverse storage effect holds for the vadose zone, that temporarily saturated conditions (as found for the hillslope scale) or groundwater influence (as found for the catchment scale) were not required to generate

younger water in recharge during periods of greater soil water storage.

The presented simulations underline that the common assumption in hydrological modelling of a well-mixed system in the subsurface does not hold for water withdrawal from the soil via evaporation, transpiration or recharge. In contrast, we saw variable water ages across the two soil pore domains and down the soil depth. Fluxes were more likely to withdraw younger water during periods of enhanced wetness and older water when the system becomes drier.

The transpiration ages shown here also indicate that water in the plant xylem have relatively old ages (and long travel times) depending on the time of the year, which is relevant for ecohydrological studies inferring root water uptake depths using stable isotopes.

**Author contribution.** M.S. conducted the simulations, made the graphs and wrote the initial manuscript; M.S., D.T., and C.S. designed the study and J.B. and H.L. provided data and site-specific knowledge for the Dorset and Krycklan

sites, respectively. All authors contributed to the writing process.

**Competing interests.** None.

**Acknowledgements**

We thank Pernilla Löfvenius (SLU) for providing PET data for Krycklan (via SITES) and Carl Mitchell for snow melt

data in Dorset. We thank Pertti Ala-aho, Paolo Benttin, Sylvain Kuppel, Aaron A. Smith, and Hailong Wang for constructive discussions on the topic. The authors would like to acknowledge the support of the Maxwell compute cluster funded by the University of Aberdeen. The Krycklan part was funded by the KAW Branch-Point project. We thank the European Research Council (ERC, project GA 335910 VeWa) for funding.





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
