# Peer review of "Water ages in the critical zone of long-term experimental sites in northern latitudes"

_Hydrology and Earth System Sciences, 2018_

## Referee Comment (RC1) · T. Walter (Referee) · 23 Apr 2018

General Comments: This study uses a previously calibrated 1-D model to ascertain estimates of travel times for different hydrological fluxes and water ages throughout the soil-plant continuum. The results generally agree with conceptual conclusions drawn from empirical studies but provide order of magnitude quantification that is hard to extract from field studies. I commend the authors for showing full distributions of travel times and water ages in their figures even though they mostly concentrate on means or medians in their narrative; I think there is some potentially interesting information in distributions that is not easily distilled into a single number. Overall, I really liked this paper and appreciated the clearly articulated short-comings, e.g., no consideration of lateral flow.

[Figure]

Specific Comments: 1) It was not clear if/how water among the different flow regimes and soil storage interacted in the model? It is possible I simply missed this detail or that it was explained in the authors' proceeding paper. 2) E and T were partitioned by vegetative cover? Was this a simple 2-d percentage over the landscape or in terms of something like leaf area index? 3) The empirical tracer experiments to which the authors compare their results are generally pretty simplistic. I encourage them to consider Kung et al. 2005. Quantifying pore-size spectrum of macropore-type preferential pathways. SSSAJ 69(4) because this empirical study used a much more complex tracer design than most studies and it sort of matches the model design used here.

Editorial Notes: 1) The first line of the abstract seems awkward; the word "respond" seems wrong. 2) I really like the use of colors in the figures but they are not always well explained (e.g., fig. 6); please make this clearer.

---

## Referee Comment (RC2) · Anonymous Referee #2 · 26 Apr 2018

This well written and structured article describes an interesting soil physical based modelling study on water travel times and water ages at four different sites in northern latitudes. The model simulations were done for an extensive period (multiple years) giving insights in both short-term and seasonal dynamics. The topic is in my opinion of interest to HESS readers and after minor revision suitable for publication. Below are my suggestions and comments for improvement of the paper:

Major comments:

1. The description of the data should be more extensive (Methods section 2.2 and 2.3). The soil hydraulic parameters used for the modelling are not mentioned in the text/table. While a reference is made to Spenger et al. 2018, having this information available in this paper really helps with the interpretation (how different are the sites for example)

[Figure]

without having to refer to Spenger et al. 2018. Furthermore, I suggest to include also other parameter values like maximum canopy storage, infiltration capacity (if applicable or a statement that overland flow does not occur). With respect to infiltration capacity; what about frozen soils at these sites?

Finally, I recommend to add some more info about the model (run), like: - What was the parameter set (it is mentioned in the paper, but not specified)? - Was there a spin up period? - What was the internal time step of the model (I guess it was forced with daily throughfall and evapotranspiration)? - Programming language, open source?

2. In section 2.4 it is not very clear to me how MTT and water ages were derived exactly. In lines 3-4 "Tracer concentrations. . .Figure 1 left)." it is mentioned that tracer concentrations in the output fluxes were normalized by the infiltrated tracer mass ($I_j(t)$). Do you mean that the mass flux (of R, T and E) was normalized by the total mass recovered at these boundaries (of R, T and E)? If so, this could be stated more clearly in my opinion (as equation?). Now it seems the normalization was done by the total infiltrated tracer mass on tracer concentrations $O_j$. This also applies to the description of the calculation of water ages. In lines 8-10 please state more clearly why MTT analysis was limited to the period 2012-2015.

3. What controls travel times and water ages in the Discussion (and Results) could be expanded to soil hydraulic parameters, for example what about differences in saturated hydraulic conductivity or saturated water content between these four sites? I strongly recommend to include these soil parameters (and advection dispersion parameters) in the analysis, since the focus of the paper is on soil physical based modelling.

4. The SWIS model solves the Richards and advection dispersion equation, and the same set of water flow and transport parameters are used for the slow and fast domain. What about possible preferential flow/ macro-pore flow at these sites, when the Richards and advection dispersion equation are probably not applicable? I recommend that the authors elaborate on this in the Discussion section.
Minor comments:

1. First sentence in the Abstract, please rewrite the part "undergo intense respond"

2. My suggestion is to move the Study sites description (2.1) from the Methods section, to a new section.

3. There seems to be a lot of overlap in the dots of Figure 2, 3 and 6. Is there a way to avoid this; different markers, make some transparent?

4. Discussion, line 4: please use references instead of "(introduced in section 1)".

5. Section 4.4 lines 2-3, what about the often observed exponential decay of root distribution with depth?

6. Section 4.2 line 9; "Due to exchange between fast and slow flow domains...", it would be good to mention in the paper on what time scale this exchange works/ how fast is this process?

7. The following publication may be of interest irt the work described in this manuscript: van Verseveld, W. J., Barnard, H. R., Graham, C. B., McDonnell, J. J., Brooks, J. R., and Weiler, M.: A sprinkling experiment to quantify celerity–velocity differences at the hillslope scale, Hydrol. Earth Syst. Sci., 21, 5891-5910, https://doi.org/10.5194/hess-21-5891-2017, 2017.

---

## Referee Comment (RC3) · Anonymous Referee #3 · 2 May 2018

This study presents interesting insights on water age dynamics in vertical soil profiles. The authors build on previous model simulations (Sprenger et al., 2017) at 4 different northern-latitude sites based on the use of a 1-D physically-based model (SWIS). While in the previous publication the authors focused on flow and isotope transport dynamics, here the focus is on the modelled age dynamics. The article is very well written and easy to follow. Results are clearly organized and fully explained. I think this manuscript will be highly appreciated by the scientific community, therefore I recommend it for publication on HESS.

In revising the manuscript, I invite the authors to consider the following comments:

1) Highlight that results are based on a model and its assumptions: All the results are based on the implementation of the SWIS model. This model was shown (Sprenger et

al., 2017) to provide reasonable soil moisture and isotope simulations. The model is evaluated on very valuable isotope data, but they only come from a single soil depth as no measurements are available at different depths or in the fluxes E, T and R. Hence, the age dynamics explored by the authors go well beyond what can be constrained by data (as typically happens in transport problems). I believe that rather different age dynamics (particularly the short-term dynamics) could likely yield equivalent model results in terms of isotope dynamics. This is fine and I do not invoke a sensitivity analysis, but keeping this uncertainty in mind, I encourage the authors to revise sentence like "Such a clear influence of vegetation on travel times" (P17L20) and to use more frequently expressions like "the model suggests that. . . " rather than "median age was. . .". Some critical discussion of the general validity of the analyses at the beginning of the discussion section would also help follow the discussion.

2) Additional insights on the SWIS model: As the paper is entirely based on the use of the SWIS model, I wonder whether further model descriptions exist that could be made available to the reader. The cited paper by Mueller et al., (2014) only includes a very short description of the model (it is just a sub-subsection of the paper!). As a reader, I came up with several questions (e.g. how does the vapour exchange simulated by the model may affect the age dynamics? How is interception modelled? How is recharge (and its age) partitioned between the different flow domains?) and it would be nice to have additional references where to find the answers.

3) Clarify the "inverse storage effect": The authors often mention the "inverse storage effect" (for example at P2L18, P14L4, P19L23) as described by Harman (2015). I think the original meaning of that terminology may have been partially misunderstood. The authors note that recharge is typically younger during higher storage periods. However, this is not enough to determine an "inverse storage effect" as recharge can be younger simply because soil water is younger (e.g. after a storm event). My understanding of what was originally intended by Harman is that during high storage conditions there are structural changes in the water transport mechanisms that lead to the activation of

faster flow pathways, ultimately causing a disproportional increase of younger water in recharge (or ET) than in the soil storage. I think the paper would benefit from improved clarity on this point.

4) Simplify the Discussion: I found the discussion section rather long and often not reflecting the section titles. For example, section 4.1 "What controls soil water storage and water ages?" includes a very large number of remarks on general storage and age dynamics (and page 15 looks like a single paragraph of 35 lines). I think the authors could improve the discussion by better focusing on: what makes this study different from existing studies on water age? What is found here that was not known before? For example, part of the discussion on the two water worlds hypothesis (P15L22-33) resembles the one already presented by Sprenger et al., 2016, Rev of Geophysics. Then, some sentences (e.g., P14L17-20 P17L3-5, P18L10-15) express results that are somewhat expected in hydrologic transport processes and could be much shortened (I think it is well established that when it rains there is younger water that infiltrates into the soil and so the soil storage becomes younger, while during dry periods soil water becomes older – and so the fluxes out of the soil).

SPECIFIC COMMENTS

Page 2, Line 5: I think a reference to earlier papers would be in place here (e.g. van der Velde 2012, Water Resour Res, Botter et al., 2010, Water Resour Res)

P2L22: I think the reference to Berghuijs and Kirchner (2017) is not in place as the paper does not discus storage variations, which are instead the crucial point in the concept of the "inverse storage effect".

P4L35: MTT usually refers to the mean transit time, so a reader that does not go through the methods will likely assume that those are mean transit times. No big deal, but I wonder if there is a more unambiguous acronym that could be used (and I am fine if the authors prefer to keep as is).

P4L34-36: I think some quick explanation on why the median is selected as travel time/age metric instead of the "traditional" mean transit time/age would be useful. The authors could specify that the median transit time (or age) is insensitive to what happens to the older component of the distribution (older than 50% of the particles). This makes the estimate more robust against the uncertainty on older water ages, but results in a "partial" metric that does not take into account the entire shape of the distribution (indeed, just the first 50%). On this, a reference to Benettin et al., 2017, Hydrol. Proc. would probably be more appropriate than Benettin et al. (2015).

P5L9: this sentence is unclear to me. To compute the median, you should only need to reach 50+% of the recovery. Instead, to compute the MTTD you need to average the entire breakthrough curves.

P5L24: technical correction: do you mean that distributions of median travel times and median water ages were derived using a cosine kernel density? I guess the age and travel time distributions were derived as described in the previous section.

Figure 5: could you show somewhere the partitioning between storage in fast flow and slow flow (maybe a figure in SI?). This would help understanding the dynamics in the total storage. Ideally it would be nice to see how E,T and R fluxes are partitioned between fast and slow domain, but I see that the article already includes many figures.

P16L17: here the authors state that "ET fluxes do not usually withdraw water from a well-mixed pool". But does this mean that the pool is not well-mixed or that ET does not withdraw water as in a well-mixed system? I think Figure 7 clearly shows that the soil water storage is not a well-mixed pool, but the problem of how the fluxes draw water out of the available soil storage is a separate problem that I think is not specifically addressed by the authors.

P17L1: is rooting depth the only difference between the two sites at Bruntland Burn? Is it possible that the different E and T fluxes could also play a difference between the two sites?

---

## Short Comment (SC1) · 4 May 2018

**Response to T. Walter (Referee 1)**

General Comments:

This study uses a previously calibrated 1-D model to ascertain estimates of travel times for different hydrological fluxes and water ages throughout the soil-plant continuum. The results generally agree with conceptual conclusions drawn from empirical studies but provide order of magnitude quantification that is hard to extract from field studies. I commend the authors for showing full distributions of travel times and water ages

in their figures even though they mostly concentrate on means or medians in their narrative; I think there is some potentially interesting information in distributions that is not easily distilled into a single number. Overall, I really liked this paper and appreciated the clearly articulated short-comings, e.g., no consideration of lateral flow.

*Response: We thank Todd Walter for taking the time to review our manuscript and for his generally positive feedback on our study.*

Specific Comments:

1) It was not clear if/how water among the different flow regimes and soil storage interacted in the model? It is possible I simply missed this detail or that it was explained in the authors' proceeding paper.

*Response: We will add and change in the methods the following sentences for clarification: " Ingraham and Criss (1993) found that two water pools approach as a function of water volumes, surface area and saturated vapor pressure (temperature) a weighted average isotopic composition of the two pools. Our previous study showed that a conceptualization of the subsurface with two pore domains that exchange water in accordance to Ingraham and Criss (1993) via the soil gas phase improved the simulation of the soil water stable isotopic composition at 10 and 20 cm depth at the investigated sites compared to an assumption of uniform flow. Therefore, we apply the same model set up of SWIS as presented in detail by Sprenger et al. (2018b) with the parameters given in Table 1. In accordance to Vanderborght and Vereecken (2007), we set the dispersivity parameter to 10 cm at all sites. The soil physical parameters were the same for the two pore domains and the exchange was solely conceptualized as vapour exchange not via hydraulic dispersion. The implemented tracer exchange between the slow and the fast flow domain results in a slow approach of the virtual tracer concentrations in the two pore domains. Thus, the exchange leads towards a homogenization of water ages between the two flow domains. In line with soil physics principles, the slow flow domain is filled first and remains saturated until the fast flow*

*domain is emptied (Hutson and Wagenet, 1995). Water flow and tracer transport occurs in both domains and recharge is generated accordingly. However, only the average recharge flux rate and weighted average tracer concentrations from both domains are provided.* ”

2) E and T were partitioned by vegetative cover? Was this a simple 2-d percentage over the landscape or in terms of something like leaf area index?

*Response: The partitioning was based on the canopy coverage, which will be now provided in a Table that lists all the parameters. We will change the sentence as follows and provide a reference: “ ET was partitioned into potential E and potential transpiration (T) according to the canopy coverage (Table 1) according to Ritchie (1972).”*

3) The empirical tracer experiments to which the authors compare their results are generally pretty simplistic. I encourage them to consider Kung et al. 2005. Quantifying pore-size spectrum of macropore-type preferential pathways. SSSAJ 69(4) because this empirical study used a much more complex tracer design than most studies and it sort of matches the model design used here.

*Response: Thanks for the suggestion. It is an interesting study and we will see if we can include it into the discussion.*

Editorial Notes:

1.) The first line of the abstract seems awkward; the word “respond” seems wrong.

*Response: We will change the sentence as follows: “As northern environments undergo intense changes. . .”*

2) I really like the use of colors in the figures but they are not always well explained (e.g., fig. 6); please make this clearer.

*Response: We will change and add in the caption of Figure 3: “The dots show the $MTT_R$ each day of rain and the colour code represents the season when the traced*

*water infiltrated the soil."* and in the caption of Figure 6: *"The dots show the relationship between water ages and storage for each day and the colour code represents the season of the corresponding days."*

---

## Short Comment (SC2) · 7 May 2018

**Response to Referee 2**

General Comments:

This well written and structured article describes an interesting soil physical based modelling study on water travel times and water ages at four different sites in northern latitudes. The model simulations were done for an extensive period (multiple years) giving insights in both short-term and seasonal dynamics. The topic is in my opinion of interest to HESS readers and after minor revision suitable for publication. Below are

[Figure]

my suggestions and comments for improvement of the paper.

*Response: We thank the anonymous Referee 2 for taking the time to review our manuscript and for their generally positive feedback on our study.*

Major Comments:

1. The description of the data should be more extensive (Methods section 2.2 and 2.3). The soil hydraulic parameters used for the modelling are not mentioned in the text/table. While a reference is made to Spenger et al. 2018, having this information available in this paper really helps with the interpretation (how different are the sites for example) without having to refer to Spenger et al. 2018. Furthermore, I suggest to include also other parameter values like maximum canopy storage, infiltration capacity (if applicable or a statement that overland flow does not occur). With respect to infiltration capacity; what about frozen soils at these sites?

*Response: We will add a new table listing the applied model parameters: Depths of the soil horizons, Mualem – van Genuchten parameters ($\theta_s$: saturated water content, $\alpha$: air entry value, n: shape parameter), saturated hydraulic conductivity K, interception capacity and canopy coverage. The infiltration capacity results from the soil hydraulic parameters. We will add the following to address soil frost: "Soil frost does usually not occur at Bruntland Burn and is rare at the Dorset site due to the insolating effect of the snow cover. At Krycklan, soil frost was shown to not induce surface runoff but soils at the forested site remained permeable (Stähli et al., 2001; Laudon et al., 2007).*

Finally, I recommend to add some more info about the model (run), like: - What was the parameter set (it is mentioned in the paper, but not specified)? - Was there a spin up period? - What was the internal time step of the model (I guess it was forced with daily throughfall and evapotranspiration)? - Programming language, open source?

*Response: We will provide the info on the temporal coverage in the first paragraph the last subsection in the methods. We will add the following sentence to include the info*

*on time steps and parameters (now listed in a table): "The model was run on daily resolution and the applied parameters are listed in Table 1." The SWIS model is written in Python code.*

2. In section 2.4 it is not very clear to me how MTT and water ages were derived exactly. In lines 3-4 "Tracer concentrations: : :Figure 1 left)." it is mentioned that tracer concentrations in the output fluxes were normalized by the infiltrated tracer mass ($I_j(t)$). Do you mean that the mass flux (of R, T and E) was normalized by the total mass recovered at these boundaries (of R, T and E)? If so, this could be stated more clearly in my opinion (as equation?). Now it seems the normalization was done by the total infiltrated tracer mass on tracer concentrations $O_j$. This also applies to the description of the calculation of water ages. In lines 8-10 please state more clearly why MTT analysis was limited to the period 2012-2015.

*Response: The tracer concentration of each flux was normalized by the total infiltrated tracer mass to get relative concentrations of each traced flow path (E, T, R). Unfortunately, we cannot follow how the Referee gets to the interpretation that we would have referred the concentrations in the flux to the total recovered mass flux in the corresponding flux. We do not see where we potentially provide unclear description with that regard as also the Referee seem have understood the procedure given the second part of their comment. Anyways, we will change the sentence for clarification: "Tracer concentrations $O_j(t)$ in the output fluxes for each day after introduction of the virtual tracer Ij at time t0 were normalized by the infiltrated tracer mass of the tracked precipitation or snowmelt event ($O_j(t)/I_j(t0)$, Figure 1 left)." To clarify that the tracer needs have entirely left the soil storage to calculate MTTs, we will change the sentence as follows: "Since MTT would be underestimated if the cumulative normalized breakthrough curve of the virtual tracers would not reach unity (tracer must have entirely left soil storage), we limited the MTT analysis to the period from 2012-2015."*

3. What controls travel times and water ages in the Discussion (and Results) could be expanded to soil hydraulic parameters, for example what about differences in saturated

hydraulic conductivity or saturated water content between these four sites? I strongly recommend to include these soil parameters (and advection dispersion parameters) in the analysis, since the focus of the paper is on soil physical based modelling.

*Response: We will add a table with the soil physical parameters and we will add the following sentence in the method's section: "In accordance to Vanderborght and Vereecken (2007), we set the dispersivity parameter to 10 cm at all sites.*

4. The SWIS model solves the Richards and advection dispersion equation, and the same set of water flow and transport parameters are used for the slow and fast domain. What about possible preferential flow/ macro-pore flow at these sites, when the Richards and advection dispersion equation are probably not applicable? I recommend that the authors elaborate on this in the Discussion section.

*Response: We will add to the discussion: "The applied model approach cannot account for preferential flow, but the conceptualization of two pore domains with different water flow and transport dynamics enabled simulating bypass flow. This conceptualization was shown to be superior to a conceptualization of a uniform flow (Sprenger et al., 2018b). Additional inclusion of preferential flow into the model domain would come on the cost of model complexity and pose problems of parameter identifiability."*

Minor comments:

1. First sentence in the Abstract, please rewrite the part "undergo intense respond"

*Response: We changed the sentence as follows: "As northern environments undergo intense changes..."*

2. My suggestion is to move the Study sites description (2.1) from the Methods section, to a new section.

*Response: We will move "Study sites" to a new section.*

3. There seems to be a lot of overlap in the dots of Figure 2, 3 and 6. Is there a way to

avoid this; different markers, make some transparent?

*Response: We will increase the transparency and make the points smaller to reduce the overlap.*

4. Discussion, line 4: please use references instead of "(introduced in section 1)".

*Response: Will be changed it as suggested*

5. Section 4.4 lines 2-3, what about the often observed exponential decay of root distribution with depth?

*Response: We assume that the Referee means that the root distribution is follows an exponential distribution. We will add the following sentence: "An exponential distribution would not change much regarding the water uptake patterns, as already the linear assumption results in 96% of the water being taken up in the upper 15 cm."*

6. Section 4.2 line 9; "Due to exchange between fast and slow flow domains: : :", it would be good to mention in the paper on what time scale this exchange works/ how fast is this process?

*Response: We will include more info on the exchange via gas phase in the methods section as follows: "Ingraham and Criss (1993) found that two water pools approach as a function of water volumes, surface area and saturated vapor pressure (temperature) a weighted average isotopic composition of the two pools. Our previous study showed that a conceptualization of the subsurface with two pore domains that exchange water in accordance to Ingraham and Criss (1993) via the soil gas phase improved the simulation of the soil water stable isotopic composition at 10 and 20 cm depth at the investigated sites compared to an assumption of uniform flow. Therefore, we apply the same model set up of SWIS as presented in detail by Sprenger et al. (2018b) with the parameters given in Table 1. In accordance to Vanderborght and Vereecken (2007), we set the dispersivity parameter to 10 cm at all sites. The soil physical parameters were the same for the two pore domains and the exchange was solely conceptualized as vapour exchange not via hydraulic dispersion. The implemented tracer exchange between the slow and the fast flow domain results in a slow approach of the virtual tracer concentrations in the two pore domains. Thus, the exchange leads towards a*

*homogenization of water ages between the two flow domains. In line with soil physics principles, the slow flow domain is filled first and remains saturated until the fast flow domain is emptied (Hutson and Wagenet, 1995). Water flow and tracer transport occurs in both domains and recharge is generated accordingly. However, only the average recharge flux rate and weighted average tracer concentrations from both domains are provided."*

7. The following publication may be of interest irt the work described in this manuscript: van Verseveld, W. J., Barnard, H. R., Graham, C. B., McDonnell, J. J., Brooks, J. R., and Weiler, M.: A sprinkling experiment to quantify celerity–velocity differences at the hillslope scale, Hydrol. Earth Syst. Sci., 21, 5891-5910, https://doi.org/10.5194/hess-21-5891-2017, 2017.

*Response: Thanks for the suggestion, we will have a close look at the given reference and see if we can include it into the discussion*

---

## Short Comment (SC3) · 7 May 2018

M. Sprenger

matthias.sprenger@abdn.ac.uk

**Response to Referee 3**

General Comments:

This study presents interesting insights on water age dynamics in vertical soil profiles. The authors build on previous model simulations (Sprenger et al., 2017) at 4 different northern-latitude sites based on the use of a 1-D physically-based model (SWIS). While in the previous publication the authors focused on flow and isotope transport dynamics, here the focus is on the modelled age dynamics. The article is very well

written and easy to follow. Results are clearly organized and fully explained. I think this manuscript will be highly appreciated by the scientific community, therefore I recommend it for publication on HESS.

*Response: We thank the anonymous Referee 3 for taking the time to review our manuscript and for their generally positive feedback on our study.*

Main Comments:

In revising the manuscript, I invite the authors to consider the following comments: 1) Highlight that results are based on a model and its assumptions: All the results are based on the implementation of the SWIS model. This model was shown (Sprenger et al., 2017) to provide reasonable soil moisture and isotope simulations. The model is evaluated on very valuable isotope data, but they only come from a single soil depth as no measurements are available at different depths or in the fluxes E, T and R. Hence, the age dynamics explored by the authors go well beyond what can be constrained by data (as typically happens in transport problems). I believe that rather different age dynamics (particularly the short-term dynamics) could likely yield equivalent model results in terms of isotope dynamics. This is fine and I do not invoke a sensitivity analysis, but keeping this uncertainty in mind, I encourage the authors to revise sentence like "Such a clear influence of vegetation on travel times" (P17L20) and to use more frequently expressions like "the model suggests that: : : " rather than "median age was: : :"Some critical discussion of the general validity of the analyses at the beginning of the discussion section would also help follow the discussion.

*Response: We will include in the revised version that the previous study was not limited to one depth, but we compared the observations and simulations for 10 and 20 cm soil depth: "Our previous study showed that such a conceptualization of the subsurface with two pore domains that exchange water in accordance to Ingraham and Criss (1993) via the soil gas phase improved the simulation of the soil water*

*stable isotopic composition at 10 and 20 cm depth at the investigated sites compared to an assumption of uniform flow." Further, we will consider rephrasing some parts as suggested by Referee 3. For their given example we will change to: "Such an influence of vegetation on travel times as suggested from the plot scale simulations is commonly not seen for the catchment runoff as the stream integrates...". We prefer to keep the critical discussion of our results in the dedicated section "Limitations and outlook", rather than splitting this section and discussing the limitations already at the beginning of the discussion.*

2) Additional insights on the SWIS model: As the paper is entirely based on the use of the SWIS model, I wonder whether further model descriptions exist that could be made available to the reader. The cited paper by Mueller et al., (2014) only includes a very short description of the model (it is just a sub-subsection of the paper!). As a reader, I came up with several questions (e.g. how does the vapour exchange simulated by the model may affect the age dynamics? How is interception modelled? How is recharge (and its age) partitioned between the different flow domains?) and it would be nice to have additional references where to find the answers.

*Response: We hope that we can clarify the open questions with the following changes: In the method's section to clarify the exchange via vapor phase and the impact on the water ages: "Ingraham and Criss (1993) found that two water pools approach as a function of water volumes, surface area and saturated vapor pressure (temperature) a weighted average isotopic composition of the two pools. Our previous study showed that a conceptualization of the subsurface with two pore domains that exchange water in accordance to Ingraham and Criss (1993) via the soil gas phase improved the simulation of the soil water stable isotopic composition at 10 and 20 cm depth at the investigated sites compared to an assumption of uniform flow. Therefore, we apply the same model set up of SWIS as presented in detail by Sprenger et al. (2018b) with the parameters given in Table 1. In accordance to Vanderborght and Vereecken (2007), we set the dispersivity parameter to 10 cm at all sites. The soil physical parameters were the same for the two pore domains and the exchange was solely conceptualized*

*as vapour exchange not via hydraulic dispersion. The implemented tracer exchange between the slow and the fast flow domain results in a slow approach of the virtual tracer concentrations in the two pore domains. Thus, the exchange leads towards a homogenization of water ages between the two flow domains. In line with soil physics principles, the slow flow domain is filled first and remains saturated until the fast flow domain is emptied (Hutson and Wagenet, 1995). Water flow and tracer transport occurs in both domains and recharge is generated accordingly. However, only the average recharge flux rate and weighted average tracer concentrations from both domains are provided."*

*To clarify the interception module: "Precipitation was divided into interception and throughfall according to the canopy coverage (Table 1), and when the interception capacity (Table 1) was reached, the surplus infiltrated into the soil."*

3) Clarify the "inverse storage effect": The authors often mention the "inverse storage effect" (for example at P2L18, P14L4, P19L23) as described by Harman (2015). I think the original meaning of that terminology may have been partially misunderstood. The authors note that recharge is typically younger during higher storage periods. However, this is not enough to determine an "inverse storage effect" as recharge can be younger simply because soil water is younger (e.g. after a storm event). My understanding of what was originally intended by Harman is that during high storage conditions there are structural changes in the water transport mechanisms that lead to the activation of faster flow pathways, ultimately causing a disproportional increase of younger water in recharge (or ET) than in the soil storage. I think the paper would benefit from improved clarity on this point.

*Response: We agree with the Referee and will revisit the use of the term "inverse storage effect". We will make clear that we refer here to the increased mobilization of water in the fast flow domain during high storage that contributed to the recharge water. However, for the E and T fluxes, flow path changes will not affect the flux ages. For these fluxes, the reduced water age in the flux stems from the increased share of young water at high storage. We will clarify these differences in the discussion,*

*while simplifying the discussion as requested below. For example we will include: "In addition to the general positive relationship between wetness and soil hydraulic conductivity (van Genuchten 1980), the conceptualization with two pore domains in the SWIS model allowed young water to bypass in the fast flow domain older water stored in the slow flow domain. Since the smaller pores of the slow flow domain will be filled first or stay filled while the bigger pores of the fast flow domain are not empty, the bypass will be enhanced during periods of high wetness."*

4) Simplify the Discussion: I found the discussion section rather long and often not reflecting the section titles. For example, section 4.1 "What controls soil water storage and water ages?" includes a very large number of remarks on general storage and age dynamics (and page 15 looks like a single paragraph of 35 lines). I think the authors could improve the discussion by better focusing on: what makes this study different from existing studies on water age? What is found here that was not known before? For example, part of the discussion on the two water worlds hypothesis (P15L22-33) resembles the one already presented by Sprenger et al., 2016, Rev of Geophysics. Then, some sentences (e.g., P14L17-20 P17L3-5, P18L10-15) express results that are somewhat expected in hydrologic transport processes and could be much shortened (I think it is well established that when it rains there is younger water that infiltrates into the soil and so the soil storage becomes younger, while during dry periods soil water becomes older – and so the fluxes out of the soil).

*Response: We agree that the discussion should be simplified and will replace the part on the basic soil hydraulic relationships with the following: "In addition to the general positive relationship between wetness and soil hydraulic conductivity (van Genuchten, 1980), the conceptualization with two pore domains in the SWIS model allowed young water to bypass in the fast flow domain older water stored in the slow flow domain. Since the smaller pores of the slow flow domain will be filled first or stay filled while the bigger pores of the fast flow domain are not empty, the bypass will be enhanced during periods of high wetness."*

*We will further delete two more sentences in the paragraph. However, we do not*

[Figure]

*agree that the section repeats already published discussion from Sprenger et al (2016, RoG), as we did not discuss the two-pore domain back than, but limited our modelling and discussion to a conceptualization with a uniform flow domain. Thus, we believe that this paragraph is quite relevant for the current discussion. We will shorten the given lines that the Referee criticized as follows: "Additionally, snowmelt led to a sharp decrease in soil water ages after a continuous aging of the water that resided in soil over the snow accumulation (Figure 5)." And we will delete and shorten the other parts the Referee referred too.*

Specific comments:

Page 2, Line 5: I think a reference to earlier papers would be in place here (e.g. van der Velde 2012, Water Resour Res, Botter et al., 2010, Water Resour Res)
*Response: We will add the references as suggested*
P2L22: I think the reference to Berghuijs and Kirchner (2017) is not in place as the paper does not discus storage variations, which are instead the crucial point in the concept of the "inverse storage effect".
*Response: Agreed, we will take the reference out*
P4L35: MTT usually refers to the mean transit time, so a reader that does not go through the methods will likely assume that those are mean transit times. No big deal, but I wonder if there is a more unambiguous acronym that could be used (and I am fine if the authors prefer to keep as is).
*Response: Even if the reader will not go through the method's section, they will still read at the beginning of each section that we talk about median travel times. We further wrote under each plot and table that we present median values. We believe therefore that we can keep the abbreviation as it is.*
P4L34-36: I think some quick explanation on why the median is selected as travel time/age metric instead of the "traditional" mean transit time/age would be useful. The

authors could specify that the median transit time (or age) is insensitive to what happens to the older component of the distribution (older than 50% of the particles). This makes the estimate more robust against the uncertainty on older water ages, but results in a "partial" metric that does not take into account the entire shape of the distribution (indeed, just the first 50%). On this, a reference to Benettin et al., 2017, Hydrol. Proc. would probably be more appropriate than Benettin et al. (2015).

*Response: We will add: "We decided to present median values, rather than mean travel times, as the latter can be biased due to uncertainties in the long tails of the transit time distributions (Seeger and Weiler, 2014)."*

P5L9: this sentence is unclear to me. To compute the median, you should only need to reach 50+% of the recovery. Instead, to compute the MTTD you need to average the entire breakthrough curves.

*Response: The recovery will be <100% if the tracer is not fully flushed out of the soil storage. We will change the sentence as follows for clarification: "Since MTT would be underestimated if the cumulative normalized breakthrough curve of the virtual tracers would not reach unity (tracer must have entirely left soil storage), we limited the MTT analysis to the period from 2012-2015."*

P5L24: technical correction: do you mean that distributions of median travel times and median water ages were derived using a cosine kernel density? I guess the age and travel time distributions were derived as described in the previous section.

*Response: Yes, we refer here to the distribution of the time variant median values and will change the sentence as follows: "Distributions of the time-variant median travel times and median water ages in fluxes and storages were derived using cosine kernel density estimations..."*

Figure 5: could you show somewhere the partitioning between storage in fast flow and slow flow (maybe a figure in SI?). This would help understanding the dynamics in the total storage. Ideally it would be nice to see how E,T and R fluxes are partitioned between fast and slow domain, but I see that the article already includes many figures.

*Response: As outlined above, ET fluxes from the slow flow domain are limited to peri-*

*ods when the fast flow domain is empty and the model output is limited to the average values for the recharge flux of both domains. However, we will add in the supplementary Figure S 4 a graph showing the soil water storage of the two flow domains.*

P16L17: here the authors state that "ET fluxes do not usually withdraw water from a well-mixed pool". But does this mean that the pool is not well-mixed or that ET does not withdraw water as in a well-mixed system? I think Figure 7 clearly shows that the soil water storage is not a well-mixed pool, but the problem of how the fluxes draw water out of the available soil storage is a separate problem that I think is not specifically addressed by the authors.

*Response: Thanks, this is a very good point and we will address this in the discussion of the revised manuscript.*

P17L1: is rooting depth the only difference between the two sites at Bruntland Burn? Is it possible that the different E and T fluxes could also play a difference between the two sites?

*Response: The differences between the forested and heather site at Bruntland Burn are not limited to the rooting depth. In addition to the rooting depth, also canopy cover and the interception storage (will be shown in a new table in the revised manuscript) affecting the infiltration volumes and the E-T partitioning, are different. We will add this also in the discussion as follows: "...our experimental set up with two different vegetation types (differing in T rates, rooting depth, canopy cover and interception storage) on similar soil types under the same climatic forcing in the Bruntland Burn..."*

---

## Referee Comment (RC4) · Anonymous Referee #2 · 11 May 2018

Thanks for the elaborate response of the authors to my comments.

Based on the Response of the authors to Major comment 2 about the derivation of MTT and water ages, the authors indicated they could not follow my interpretation that "we would have referred the concentrations in the flux to the total recovered mass flux in the corresponding flux". The authors also state that "The tracer concentration of each flux was normalized by the total infiltrated tracer mass to get relative concentrations of each traced flow path (E, T, R)."

However, if I understand correctly we're dealing here with unsteady-state water flow conditions, right? In that case you should calculate mass fluxes (tracer concentration x flux (of E or T or R)), and normalize by the total recovered mass (of each traced flow path). Normalizing tracer concentrations is only possible for steady-state water flow

conditions.

It's well possible I missed some detail in the manuscript that explains the reasoning for normalizing tracer concentrations, or that I don't understand the approach/methodology completely, however could you please explain this?

---

## Short Comment (SC4) · 16 May 2018

We thank Referee 2 for their response to our comments, which clarified their issue raised.

We agree with Referee 2 that we are dealing with unsteady-state flow conditions and that we are using mass fluxes of E, T and R to calculate the median travel times. Since we define the median as the time half of the cumulative mass flux was passed the flow, the median travel time calculations do not need to be normalized. We will clarify that in the revised version.

However, for the calculations of the water ages in the fluxes and the storage, we need to know how much water of individual infiltration events (days of rainfall or snowmelt) is on each day of the simulation in the flux or storage of interest. Therefore, we need

to relate the concentration of the virtual tracer in the flux (or storage), $O_J$(t), to the total mass introduced of the virtual tracer, $I_J$($t_0$). Note that we introduce for each day of rainfall or snowmelt an individual virtual tracer (with $I_J$, $I_{J+1}$, $I_{J+2}$, $I_{J+3}$, ... being the tracer on the first, second, and third day of infiltrating water, respectively). Thus, in this case, we need to do the normalization ($O_J$(t) / $I_J$($t_0$)), which was criticized by Referee 2.

Maybe Referee 2 can provide references to show why this normalization is not valid under non-steady-state conditions.

---

## Referee Comment (RC5) · Anonymous Referee #2 · 17 May 2018

Thanks for the response and explanation.

My main criticism is that for the derivation of MTT and Water Age, under unsteady-state flow conditions, tracer **concentrations** $Oj(t)$ are used, while **mass fluxes** should be used. Just for clarity, my interpretation of mentioned tracer concentrations $Oj(t)$ in the Manuscript, is of actual tracer concentration ($[gl^{-1}]$) in the output fluxes. In case of unsteady-state flow conditions mass fluxes, and not concentrations, should be used for the calculation of MTT and Water Age.

With regard to derivation of MTT:

The authors agree that unsteady-state conditions occur and mentiond in their response "that we are using mass fluxes of E, T and R to calculate the median travel times".

[Figure]

Does this mean:

1) the authors will re-calculate the median travel times? or

2) mass fluxes were used already for the analysis of MTT?

The reason for asking this is that the authors state on page 5, lines 1-4, in the Manuscript that tracer concentrations were used, specifically in lines 3-4: "Tracer concentrations $O_j(t)$ in the output fluxes for each day after introduction of the virtual tracer $I_j$ at time $t_0$ were normalized by the infiltrated tracer mass $(O_j(t)/I_j(t0)$, Figure 1 left)."

Furhermore the authors proposed in their first repsonse to change this sentence to:

"Tracer concentrations $O_j(t)$ in the output fluxes for each day after introduction of the virtual tracer $I_j$ at time $t_0$ were normalized by the infiltrated tracer mass of the tracked precipitation or snowmelt event $(O_j(t)/I_j(t_0)$, Figure 1 left)."

In my opinion this should be changed to for example:

"Tracer mass fluxes $M_j(t)$ in the outputs for each day after introduction of the virtual tracer $I_j$ at time $t_0$ were normalized by the total recovered mass $M_t$ of the output fluxes $(M_j(t)/M_t$, Figure 1 left)."

mass flux is calculated then as follows:

$Massflux(t) = O_j(t) \cdot flux$

with flux for example in $[ls^{-1}]$

Indeed as the authors state in their response, normalization is not required here, since the median travel time is defined as the time half of the cumulative mass flux was passed the flow. In addition, a PDF with unity of one is not required for their analysis (reason why one would normalize by the total recovered mass in the output).

With regard to derivation of Water Age:

As for the derivation of MTT my main concern here is the use of tracer concentrations

$O_j(t)$, during unsteady-state flow conditions, for the calculation of water ages. Because of the unsteady-state flow conditions, you need to include a flow rate (E, T and R) or volume in case of storage. This also means,at least for E, T and R you cannot normalize by the tracer mass inputs, since not all input tracer mass is recovered at these flux boundaries.

———————————————————

---

## Referee Comment (RC6) · Anonymous Referee #2 · 18 May 2018

Dear Matthias, other authors,

I already thought there might be some confusion about the used terms. Good to read that this is solved now, and that the analysis was done in the correct way!

My last (and earlier raised) point is the possible issue with the calculation of Water Ages, based on normalization by using the Inflow Mass. Normalizing by Inflow Mass is of course fine when there is one outflow boundary, where all (or most) Inflow Mass is recovered. I am not so sure that this works for multiple outflow boundaries (R, E and T), with unsteady-state flow conditions.

Let's say you want to calculate the mean travel time at one of the outflow boundaries (R, E, and T), of an introduced virtual tracer $I_j$, then you would normalize the mass flux

by the recovered mass at that outflow boundary (to get a travel time distribution, and to calculate the mean travel time from this distribution), and you would not normalize by the Inflow tracer mass $I_j$. Right? I think the approach of normalizing by recovered mass tracer should also be applied to the calculation of Water Ages. In addition, since unsteady-state flow conditions occur, the weighting by Inflow Mass is not consistent for different tracer inflow events $I_j$, simply because the ratio of recovered mass to inflow mass $I_j$ is not equal for each inflow mass event $I_j, I_j + 1, I_j + 2, ....$

---

## Short Comment (SC5) · 18 May 2018

Dear Referee 2,
I am very thankful for your time and patience to guarantee the quality of the study. You made your point very clear, I know understand your concern and I have to apologize for not having stated the methods correctly. Though, I should have known better, since I had a similar comment during the revision of my paper in WRR in 2016, using a similar approach. We would have saved quite some time if I would have been clearer in the manuscript, sorry.

All travel times and water ages that we present in the originally submitted manuscript were based on mass fluxes, where the concentration of the virtual tracer (Mass/Volume) in a flux was multiplied by the flow (Volume/Time), leading to a mass

flux (Mass/Time). That I wrote about concentrations is misleading and does not reflect what was done in the analysis.

This will be clearly stated in the revised manuscript and in line with the suggestions made by Referee 2, the labeling in Figure 1 will be changed accordingly (see attached Figure). As suggested we will use letter "$M$", representing "Mass flux" and not "$O$", originally supposed to mean "Outflow" (in contrast to "$I$", representing "Inflow").

[Figure]

[Figure]

**Median travel time** (*MTT*): time until 50 % of the virtual tracer ($I_j$) passed in the output flux

**Median water age** (*A*): age of the 0.5 percentile in the age distribution in outflow flux or storage

**Fig. 1.** Revised Figure 1 with changed labels.

---

## Referee Comment (RC7) · Anonymous Referee #2 · 22 May 2018

Thanks for the detailed explanation, looks good!

---

## Short Comment (SC6) · 22 May 2018

Calculation of mass flux normalized by the infiltrated mass O(t):

$O(t) = [ C(t) * Q(t) ] / I(t=0)$          Unit: $[g/l * l/s / kg = 1/s]$

With $C(t)$ [g/l] is the concentration in the flux, $Q(t)$ [l/s] is the flux, $I(t=0)$ [kg]

[Figure]

In the cumulative distribution of O(t), one can see in this example, that about 51% of the introduced tracer left the soil via this outflow (i.e.g, recharge). The median travel time was calculated as the median of the 51% in this case (i.e., at 20.5% here).

[Figure]

To get the amount V(t) [mm/s] of precipitation in the flux, multiply with the introduced volume, P(t=0):

$V(t) = O(t) * P(t=0)$          Unit: [mm/s]

[Figure]

To get the share of this precipitation event of the total flux Vr(t), divide by the flux volume on the day of interest F(t) [mm]:

Vr(t) = V(t) / F(t)                    Unit: 1 / day

[Figure]

When Vr(t) is summarized over all virtual tracers, one can see that all water in the flux is eventually composed of the virtual tracers: It approaches unity when all water, that was stored initially in the soil (of unknown age) has been recharged.

[Figure]

We can then add the share of water in the flux of different ages and get a cumulative age distribution. The median of that distribution (0.5 on y-axis) shows the day of infiltrated water of which half in the flux is older than that day and the other half in the flux is younger. The water age is then difference between this day and the day of interest in the flux.

---

## Author Comment (AC1) · 24 May 2018

Please see SC1 for detailed response by the authors to the comments made by Referee #1.

———————————————————

---

## Author Comment (AC2) · 24 May 2018

Please see SC2, SC4, SC5, and SC6 for detailed responses by the authors to the comments made by Referee #2. We will address the issues raised by Referee #2 in the revised manuscript to ensure that the procedure of how we derived median travel times and median water ages is can be understood by the reader.
* * *

---

## Author Comment (AC3) · 24 May 2018

Please see SC3 for detailed response by the authors to the comments made by Referee #3.

---

## Author Response (AR1)

**General Response to the Editor and Referees**

*We thank the Editor and the three Referees for their time and effort to ensure high quality of our study and to help clarifying issues that came up reading the methods section. We appreciate the public discussion and are glad that we could resolve the issues raised by the Referees in that process. We included all the comments into the revised manuscript and you can follow the changes, since they were tracked and highlighted.*

*We prepared a response to each comment given by the editors. These responses are shown below in blue italic font and usually include citation in quotation marks that show the changed or added sentences.*

*Given the successful open discussion in HESSD and the general positive responses regarding the significance and quality in the Referee reports, we are hopeful that the handling Editor will accept the revised manuscript for publication in HESS.*

*Since all Referees had questions regarding the methods section, we would like to highlight here two main changes, where we added info on the two-pore domain and the exchange via vapor phase, and the calculation of the median travel times and water ages.*

*We changed and added the following to clarify the vapor exchange between the fast and the slow flow domain: "Ingraham and Criss (1993) found that two water pools approach as a function of water volumes, surface area and saturated vapor pressure (temperature) a weighted average isotopic composition of the two pools. Our previous study showed that a conceptualization of the subsurface with two pore domains that exchange water in accordance with Ingraham and Criss (1993) via the soil gas phase improved the simulation of the soil water stable isotopic composition at 10 and 20 cm depths at the investigated sites compared to an assumption of uniform flow. Therefore, we apply the same model set up of SWIS as presented in detail by Sprenger et al. (2018) with the parameters given in Table 1. In accordance with Vanderborght and Vereecken (2007), we set the dispersivity parameter to 10 cm at all sites. The soil physical parameters were the same for the two pore domains and the exchange was solely conceptualized as vapour exchange rather than via hydraulic dispersion. The implemented tracer exchange between the slow and the fast flow domain results in a slow approach of the virtual tracer concentrations in the two pore domains. Thus, the exchange leads towards a homogenization of water ages between the two flow domains. Consistent with soil physics principles, the slow flow domain is filled first and remains saturated until the fast flow domain is emptied (Hutson and Wagenet 1995). Water flow and tracer transport occurs in both domains and recharge is generated accordingly. However, only the average recharge flux rate and weighted average tracer concentrations from both domains are provided."*

*We further added the following Table to list the model parameters:*

*Table 1 Model parameters: Depths of the soil horizons, Mualem – van Genuchten parameters ($\vartheta_r$: residual water content, $\vartheta_s$: saturated water content, $\alpha$: air entry value, n: shape parameter), saturated hydraulic conductivity K, interception capacity and canopy coverage (Sprenger et al., 2018).*

| Site | Depth [cm] | $\vartheta_r$ [cm³ cm⁻³] | $\vartheta_s$ [cm³ cm⁻³] | $\alpha$ [cm⁻¹] | n [-] | K [cm d⁻¹] | Interception capacity [mm] | Canopy coverage [%] |
|---|---|---|---|---|---|---|---|---|
| Bruntland Burn, forested | 0-15 | 0.0454 | 0.6048 | 0.0434 | 1.3680 | 345.18 | 7.5 | 63 |
| | 15-50 | 0.0375 | 0.4936 | 0.0422 | 1.4542 | 322.89 | | |
| Bruntland Burn, heather | 0-15 | 0.0415 | 0.5822 | 0.0431 | 1.3765 | 392.46 | 2.65 | 60 |
| | 15-50 | 0.0387 | 0.4435 | 0.0452 | 1.7185 | 282.54 | | |
| Dorset | 0-25 | 0.0456 | 0.6082 | 0.0221 | 1.3672 | 485.04 | 2.2 | 89 |
| | 25-50 | 0.0356 | 0.5136 | 0.0238 | 1.3937 | 427.09 | | |

| | | | | | | | | |
|---|---|---|---|---|---|---|---|---|
| *Krycklan* | *0-20* | *0.0429* | *0.70* | *0.0919* | *1.4895* | *147†* | *1.3* | *95* |
| | *20-50* | *0.0472* | *0.5* | *0.0835* | *1.7469* | *656†* | | |

*We revised the description of the calculation of the median travel times as follows:*

*"To derive the MdTT, we extracted the breakthrough curves of the normalized mass fluxes $O_j(t)$ in the output fluxes (E, T, and R) generated from each virtual tracer mass introduced during individual infiltration events ($I_j$) on day j (Figure 1 left). Normalized mass fluxes $O_j(t)$ [$T^{-1}$] resulted from the tracer concentration (introduced with $I_j$) in the flux $C_j(t)$ [$M* L^{-3}$] times flux rate $Q(t)$ [$L^3*T^{-1}$] divided by the introduced tracer mass $I_j$ [M]:*

$$O_j(t) = \frac{C_j(t)*Q(t)}{I_j(t=0)} \qquad Eq.\ (1)$$

*We then computed the median of the individual breakthrough curves as half of the maximum cumulative $O_j(t)$, which then described the time it took until 50 % of the infiltrated water ended up in the considered output flux from the soil."*

*We revised the description of the median water ages as follows:*

*"We calculated water ages, $A(t_i)$ by first multiplying $O_j(t)$ [$T^{-1}$] with the precipitation amount $P_j(t=0)$ [$L^3$] that introduced the virtual tracer $I_j(t=0)$ and divided it by the flux volume $Q(t_i)$ on the day that we estimated the water ages for, to get the relative share of each precipitation event introduced on day $t_j$ in the considered fluxes $V_j(t)$ [$T^{-1}$].*

$$V_j(t) = O_j(t) * \frac{P(t_j)}{Q(t_i)} \qquad Eq.\ (2)$$

*Multiplication of $V_j$ with the days since $t_j$ provides the relative volume of the water of age $t_i$-$t_j$ for each considered day (Figure 1 right). The 50th percentile of the cumulative sum of $V_j$ then defined the median water age. To prevent bias due to water of unknown age in the soil storage (i.e., initial water in the soil at start of simulation), we limited the water age analysis to the period from 2013-2016."*

**Todd Walter, Referee #1**

**General Comments**

This study uses a previously calibrated 1-D model to ascertain estimates of travel times for different hydrological fluxes and water ages throughout the soil-plant continuum. The results generally agree with conceptual conclusions drawn from empirical studies but provide order of magnitude quantification that is hard to extract from field studies. I commend the authors for showing full distributions of travel times and water ages in their figures even though they mostly concentrate on means or medians in their narrative; I think there is some potentially interesting information in distributions that is not easily distilled into a single number. Overall, I really liked this paper and appreciated the clearly articulated short-comings, e.g., no consideration of lateral flow.

Response: We thank Todd Walter for taking the time to review our manuscript and for his generally positive feedback on our study.

**Specific Comments**

1) It was not clear if/how water among the different flow regimes and soil storage interacted in the model? It is possible I simply missed this detail or that it was explained in the authors' proceeding paper.

*Response: Please see the general response and the given changes in the methods section.*

2) E and T were partitioned by vegetative cover? Was this a simple 2-d percentage over the landscape or in terms of something like leaf area index?

*Response: The partitioning was based on the canopy coverage, which is provided in the revised manuscript in a Table that lists all the parameters. We changed the sentence as follows and provide a reference: " ET was partitioned into potential E and potential transpiration (T) according to the canopy coverage (Table 1) according to Ritchie (1972)."*

3) The empirical tracer experiments to which the authors compare their results are generally pretty simplistic. I encourage them to consider Kung et al. 2005. Quantifying pore-size spectrum of macropore-type preferential pathways. SSSAJ 69(4) because this empirical study used a much more complex tracer design than most studies and it sort of matches the model design used here.

*Response: Thanks for highlighting this manuscript. We had a close look at the suggested study and found in Hutson and Wagenet (1995) a very relevant reference for our manuscript. However, we did not include the originally suggested reference Kung et al. (2005), since their study focusses on preferential flow simulations, which our modelling approach does not cover in a comparable way.*

**Editorial Notes**

1) The first line of the abstract seems awkward; the word "respond" seems wrong.

*Response: We changed the sentence as follows: "As northern environments undergo intense changes…"*

2) I really like the use of colors in the figures but they are not always well explained (e.g., fig. 6); please make this clearer.

*Response: We changed and added in the caption of Figure 3: "The dots show the MTT$_R$ for each day of rain and the colour code represents the season when the traced water infiltrated the soil." and in the caption of Figure 6: "The dots show the relationship between water ages and storage for each day and the colour code represents the season of the corresponding days."*

**Referee #2**

**General Comments**

This well written and structured article describes an interesting soil physical based modelling study on water travel times and water ages at four different sites in northern latitudes. The model simulations were done for an extensive period (multiple years) giving insights in both short-term and seasonal dynamics. The topic is in my opinion of interest to HESS readers and after minor revision suitable for publication. Below are my suggestions and comments for improvement of the paper.

*Response: We thank the anonymous Referee 2 for taking the time to review our manuscript and for their generally positive feedback on our study.*

**Major Comments**

1. The description of the data should be more extensive (Methods section 2.2 and 2.3). The soil hydraulic parameters used for the modelling are not mentioned in the text/table. While a reference is made to Spenger et al. 2018, having this information available in this paper really helps with the interpretation (how different are the sites for example) without having to refer to Spenger et al. 2018. Furthermore, I suggest to include also other parameter values like maximum canopy storage, infiltration capacity (if applicable or a statement that overland flow does not occur). With respect to infiltration capacity; what about frozen soils at these sites?

*Response: We added a new table (Table 1 in revised manuscript) listing the applied model parameters: Depths of the soil horizons, Mualem – van Genuchten parameters ($\vartheta_r$: residual water content, $\vartheta_s$: saturated water content, α: air entry value, n: shape parameter), saturated hydraulic conductivity K, interception capacity and canopy coverage. The infiltration capacity results from the soil hydraulic parameters. We added the following to address soil frost: "Soil frost does usually not occur at Bruntland Burn and is rare at the Dorset site due to the insulating effect of the snow cover. At Krycklan, soil frost was shown to not*

*induce surface runoff but soils at the forested site remained permeable (Stähli et al. 2001; Laudon et al. 2007).*

Finally, I recommend to add some more info about the model (run), like: - What was the parameter set (it is mentioned in the paper, but not specified)? - Was there a spin up period? - What was the internal time step of the model (I guess it was forced with daily throughfall and evapotranspiration)? - Programming language, open source?

*Response: We provide now the info on the temporal coverage in the first paragraph in the last subsection in the methods. We added the following sentence to include the info on time steps and parameters (now listed in a table): "The model was run at daily resolution and the applied parameters are listed in Table 1." The SWIS model is written in Python code.*

2. In section 2.4 it is not very clear to me how MTT and water ages were derived exactly. In lines 3-4 "Tracer concentrations: : :Figure 1 left)." it is mentioned that tracer concentrations in the output fluxes were normalized by the infiltrated tracer mass (Ij(t)). Do you mean that the mass flux (of R, T and E) was normalized by the total mass recovered at these boundaries (of R, T and E)? If so, this could be stated more clearly in my opinion (as equation?). Now it seems the normalization was done by the total infiltrated tracer mass on tracer concentrations Oj. This also applies to the description of the calculation of water ages. In lines 8-10 please state more clearly why MTT analysis was limited to the period 2012-2015.

*Response: We thank the Referee #2 for their detailed questions on this matter and the discussion we had on HESSD (see RC2, SC2, RC4, SC4, RC5, SC5, RC6 and SC6).*
*We refer to the above given changes and additions for the methods section.*

3. What controls travel times and water ages in the Discussion (and Results) could be expanded to soil hydraulic parameters, for example what about differences in saturated hydraulic conductivity or saturated water content between these four sites? I strongly recommend to include these soil parameters (and advection dispersion parameters) in the analysis, since the focus of the paper is on soil physical based modelling.

*Response: We added a table with the soil physical parameters and we added the following sentence in the methods section: "In accordance to Vanderborght and Vereecken (2007), we set the dispersivity parameter to 10 cm at all sites."*
*We included the hydraulic conductivity in the discussion as follows: "According to our simulations, water ages are not simply controlled by the hydraulic conductivity of the soil, but the storage dynamics in the slow and fast domain also impacted the water age dynamics. Water flow was much slower when the fast flow domain emptied. Consequently, while the hydraulic conductivities at the Bruntland Burn and Dorset sites were similar (Table 1), the water ages at the two forested sites, where the fast flow domain dried out during summer, were greater than at the heather site in at Bruntland Burn, where water flow prevailed in the fast flow domain throughout the year."*

4. The SWIS model solves the Richards and advection dispersion equation, and the same set of water flow and transport parameters are used for the slow and fast domain. What about possible preferential flow/ macro-pore flow at these sites, when the Richards and advection dispersion equation are probably not applicable? I recommend that the authors elaborate on this in the Discussion section.

*Response: We added to the discussion: "The applied model approach cannot account for preferential flow, but the conceptualization of two pore domains with different water flow and transport dynamics enabled simulation of bypass flow. This conceptualization was shown to be superior to a conceptualization of a uniform flow (Sprenger et al., 2018b). Additional inclusion of preferential flow into the model domain would come at the cost of model complexity and pose problems of parameter identifiability."*

Minor comments

1. First sentence in the Abstract, please rewrite the part "undergo intense respond"

*Response: We changed the sentence as follows: "As northern environments undergo intense changes…:"*
2. My suggestion is to move the Study sites description (2.1) from the Methods section, to a new section.
*Response: "Study sites" is now its own section (Section 2 in the revised manuscript).*
3. There seems to be a lot of overlap in the dots of Figure 2, 3 and 6. Is there a way to avoid this; different markers, make some transparent?
*Response: We increased the transparency and made the points smaller to reduce the overlap for Figure 2, 3 and 6.*

[Figure]

*Revised Figure 2: Note that marker size was reduced and transparency was increased.*

[Figure]

*Revised Figure 3: Note that the markers are smaller and transparency is increased in this revised figure.*

[Figure]

*Revised Figure 6: Note that the markers are smaller and transparency was increased in this revised figure.*

4. Discussion, line 4: please use references instead of "(introduced in section 1)".

*Response: Changed it as suggested*

5. Section 4.4 lines 2-3, what about the often observed exponential decay of root distribution with depth?

*Response: We assume that the Referee means that the root distribution follows an exponential distribution. We added the following sentence: "An exponential distribution would not change the water uptake patterns significantly, as the linear assumption already results in 96% of the water being taken up in the upper 15 cm."*

6. Section 4.2 line 9; "Due to exchange between fast and slow flow domains: : :", it would be good to mention in the paper on what time scale this exchange works/ how fast is this process?

*Response: Please see the changes and additional info provided above in the General Response.*

7. The following publication may be of interest irt the work described in this manuscript:

van Verseveld, W. J., Barnard, H. R., Graham, C. B., McDonnell, J. J., Brooks, J. R., and Weiler, M.: A sprinkling experiment to quantify celerity–velocity differences at the hillslope scale, Hydrol. Earth Syst. Sci., 21, 5891-5910, https://doi.org/10.5194/hess-21-5891-2017, 2017.

*Response: We thank Referee #2 for their suggestion of looking into van Versefeld et al.. However, after reading the study, we believe that including hillslope experiments would go beyond the current discussion, as we focus on plot scale simulations.*

**Referee 3**

**General Comments**

This study presents interesting insights on water age dynamics in vertical soil profiles. The authors build on previous model simulations (Sprenger et al., 2017) at 4 different northern-latitude sites based on the use of a 1-D physically-based model (SWIS). While in the previous publication the authors focused on flow and isotope transport dynamics, here the focus is on the modelled age dynamics. The article is very well written and easy to follow. Results are clearly organized and fully explained. I think this manuscript will be highly appreciated by the scientific community, therefore I recommend it for publication on HESS.

*Response: We thank the anonymous Referee 3 for taking the time to review our manuscript and for their generally positive feedback on our study.*

**Main Comments**

In revising the manuscript, I invite the authors to consider the following comments: 1) Highlight that results are based on a model and its assumptions: All the results are based on the implementation of the SWIS model. This model was shown (Sprenger et al., 2017) to provide reasonable soil moisture and isotope simulations. The model is evaluated on very valuable isotope data, but they only come from a single soil depth as no measurements are available at different depths or in the fluxes E, T and R. Hence, the age dynamics explored by the authors go well beyond what can be constrained by data (as typically happens in transport problems). I believe that rather different age dynamics (particularly the short-term dynamics) could likely yield equivalent model results in terms of isotope dynamics. This is fine and I do not invoke a sensitivity analysis, but keeping this uncertainty in mind, I encourage the authors to revise sentence like "Such a clear influence of vegetation on travel times" (P17L20) and to use more frequently expressions like "the model suggests that: : :" rather than "median age was: : :"Some critical discussion of the general validity of the analyses at the beginning of the discussion section would also help follow the discussion.

*Response: We included in the revised version that the previous study was not limited to one depth, but we compared the observations and simulations for 10 and 20 cm soil depths: "Our previous study showed that such a conceptualization of the subsurface with two pore domains that exchange water in accordance to Ingraham and Criss (1993) via the soil gas phase improved the simulation of the soil water stable isotopic composition at 10 and 20 cm depths at the investigated sites compared to an assumption of uniform flow."*

*Further, we rephrased several parts as suggested by Referee 3. For their given example we changed the sentence to: "Such an influence of vegetation on travel times as suggested from the plot scale simulations is commonly not seen for the catchment runoff as the stream integrates…:". However, we prefer to keep the critical discussion of our results in the dedicated section "Limitations and outlook", rather than splitting this section and discussing the limitations already at the beginning of the discussion.*

2) Additional insights on the SWIS model: As the paper is entirely based on the use of the SWIS model, I wonder whether further model descriptions exist that could be made available to the reader. The cited paper by Mueller et al., (2014) only includes a very short description of the model (it is just a sub-subsection of the paper!). As a reader, I came up with several questions (e.g. how does the vapour exchange simulated by the model may affect the age dynamics? How is interception modelled? How is recharge (and its age) partitioned between the different flow domains?) and it would be nice to have additional references where to find the answers.

*Response: Please see our General Response above regarding additional info on the vapour exchange.*

*To clarify the interception module we added: "Precipitation was divided into interception and throughfall according to the canopy coverage (Table 1), and when the interception capacity (Table 1) was reached, the surplus infiltrated into the soil."*

3) Clarify the "inverse storage effect": The authors often mention the "inverse storage effect" (for example at P2L18, P14L4, P19L23) as described by Harman (2015). I think the original meaning of that terminology may have been partially misunderstood. The authors note that recharge is typically younger during higher storage periods. However, this is not enough to determine an "inverse storage effect" as recharge can be younger simply because soil water is younger (e.g. after a storm event). My understanding of what was originally intended by Harman is that during high storage conditions there are structural changes in the water transport mechanisms that lead to the activation of faster flow pathways, ultimately causing a disproportional increase of younger water in recharge (or ET) than in the soil storage. I think the paper would benefit from improved clarity on this point.

*Response: We agree with the Referee and revisited the use of the term "inverse storage effect". We clarified that we refer here to the increased mobilization of water in the fast flow domain during high storage that contribute to recharge water. However, for the E and T fluxes, flow path changes will not affect the flux ages. For these fluxes, the reduced water age in the flux stems from the increased share of young water at high storage. We clarified these differences in the discussion, while simplifying the discussion as requested below. For example we included:*

*"In addition to the general positive relationship between wetness and soil hydraulic conductivity (van Genuchten 1980), the conceptualization with two pore domains in the SWIS model allowed young water in the fast flow domain to bypass older water stored in the slow flow domain. Since the smaller pores of the slow flow domain will be filled first or stay filled while the larger pores of the fast flow domain are not empty, the bypass will be enhanced during periods of high wetness."*

4) Simplify the Discussion: I found the discussion section rather long and often not reflecting the section titles. For example, section 4.1 "What controls soil water storage and water ages?" includes a very large number of remarks on general storage and age dynamics (and page 15 looks like a single paragraph of 35 lines). I think the authors could improve the discussion by better focusing on: what makes this study different from existing studies on water age? What is found here that was not known before? For example, part of the discussion on the two water worlds hypothesis (P15L22-33) resembles the one already presented by Sprenger et al., 2016, Rev of Geophysics. Then, some sentences (e.g., P14L17-20 P17L3-5, P18L10-15) express results that are somewhat expected in hydrologic transport processes and could be much shortened (I think it is well established that when it rains there is younger water that infiltrates into the soil and so the soil storage becomes younger, while during dry periods soil water becomes older – and so the fluxes out of the soil).

*Response: We agree that the discussion should be simplified and thus replaced, for example, the part on the basic soil hydraulic relationships with the quoted sentence in the response to the last comment of the Referee. We further deleted two more sentences in the paragraph. However, we do not agree that the section repeats already published discussion from Sprenger et al (2016, RoG), as we did not discuss the two-pore domain in that paper, but limited our modelling and discussion to a conceptualization with a uniform flow domain for that manuscript. Thus, we believe that this paragraph is quite relevant for the current discussion. We have shortened the given lines that the Referee criticized as follows: "Additionally, snowmelt led to a sharp decrease in soil water ages after a continuous aging of the water that resided in soil over the snow accumulation period (Figure 5)." And we have deleted and shortened the other parts of the Discussion that the Referee referred to.*

**Specific comments**

Page 2, Line 5: I think a reference to earlier papers would be in place here (e.g. van der Velde 2012, Water Resour Res, Botter et al., 2010, Water Resour Res)

*Response: We added the references as suggested*

P2L22: I think the reference to Berghuijs and Kirchner (2017) is not in place as the paper does not discus storage variations, which are instead the crucial point in the concept of the "inverse storage effect".

*Response: Agreed, we have removed the reference.*

P4L35: MTT usually refers to the mean transit time, so a reader that does not go through the methods will likely assume that those are mean transit times. No big deal, but I wonder if there is a more unambiguous acronym that could be used (and I am fine if the authors prefer to keep as is).

*Response: We changed the abbreviation as suggested and name it now "MdTT" in line with, for example, Rodriguez et al. (2018). The change was done for the text, the tables and the figures.*

P4L34-36: I think some quick explanation on why the median is selected as travel time/age metric instead of the "traditional" mean transit time/age would be useful. The authors could specify that the median transit time (or age) is insensitive to what happens to the older component of the distribution (older than 50% of the particles). This makes the estimate more robust against the uncertainty on older water ages, but results in a "partial" metric that does not take into account the entire shape of the distribution (indeed, just the first 50%). On this, a reference to Benettin et al., 2017, Hydrol. Proc. would probably be more appropriate than Benettin et al. (2015).

*Response: We added: "We decided to present median values, rather than mean travel times, as the latter can be biased due to uncertainties in the long tails of the transit time distributions (Seeger and Weiler, 2014)."*

P5L9: this sentence is unclear to me. To compute the median, you should only need to reach 50+% of the recovery. Instead, to compute the MTTD you need to average the entire breakthrough curves.

*Response: The recovery will be <100% if the tracer is not fully flushed out of the soil storage. We will change the sentence as follows for clarification: "Since MdTT would be underestimated if the cumulative normalized breakthrough curve of the virtual tracers would not reach unity (tracer must have entirely left soil storage), we limited the MdTT analysis to the period from 2012-2015."*

P5L24: technical correction: do you mean that distributions of median travel times and median water ages were derived using a cosine kernel density? I guess the age and travel time distributions were derived as described in the previous section.

*Response: Yes, we refer here to the distribution of the time variant median values and changed the sentence as follows: "Distributions of the time-variant median travel times and median water ages in fluxes and storages were derived using cosine kernel density estimations…"*

Figure 5: could you show somewhere the partitioning between storage in fast flow and slow flow (maybe a figure in SI?). This would help understanding the dynamics in the total storage. Ideally it would be nice to see how E,T and R fluxes are partitioned between fast and slow domain, but I see that the article already includes many figures.

*Response: As outlined above, ET fluxes from the slow flow domain are limited to periods when the fast flow domain is empty and the model output is limited to the average values for the recharge flux of both domains. However, we added in the supplementary Figure S 4 a graph showing the soil water storage of the two flow domains.*

P16L17: here the authors state that "ET fluxes do not usually withdraw water from a well-mixed pool". But does this mean that the pool is not well-mixed or that ET does not withdraw water as in a well-mixed system? I think Figure 7 clearly shows that the soil water storage is not a well-mixed pool, but the problem of how the fluxes draw water out of the available soil storage is a separate problem that I think is not specifically addressed by the authors.

*Response: Thanks, we address this in the discussion of the revised manuscript, for example here:*
*"In particular, our simulations underline that ET flux withdrawal is neither well mixed in its age composition nor is the pool of plant water uptake well-mixed, which is increasingly acknowledged in water age studies (Harman 2015; Smith et al. 2018)."*

P17L1: is rooting depth the only difference between the two sites at Bruntland Burn? Is it possible that the different E and T fluxes could also play a difference between the two sites?

*Response: The differences between the forested and heather site at Bruntland Burn are not limited to the rooting depth. In addition to the rooting depth, canopy cover and interception storage, which are now listed in Table 1 of the revised manuscript, also differ. These differences affect the infiltration volumes and the E-T partitioning leading to different water balances and fluxes. We added this in the discussion as follows: "...our experimental set up with two different vegetation types (differing in T rates, rooting depth, canopy cover and interception storage) on similar soil types under the same climatic forcing in the Bruntland Burn..."*

**Publication bibliography**

Harman, Ciaran J. (2015): Time-variable transit time distributions and transport: Theory and application to storage-dependent transport of chloride in a watershed. In *Water Resour. Res.* 51 (1), pp. 1–30. DOI: 10.1002/2014WR015707.

Hutson, J. L.; Wagenet, R. J. (1995): A Multiregion Model Describing Water Flow and Solute Transport in Heterogeneous Soils. In *Soil Science Society of America Journal* 59 (3), p. 743. DOI: 10.2136/sssaj1995.03615995005900030016x.

Ingraham, Neil L.; Criss, Robert E. (1993): Effects of surface area and volume on the rate of isotopic exchange between water and water vapor. In *J. Geophys. Res.* 98 (D11), pp. 20547–20553. DOI: 10.1029/93JD01735.

Laudon, Hjalmar; Sjöblom, Viktor; Buffam, Ishi; Seibert, Jan; Mörth, Magnus (2007): The role of catchment scale and landscape characteristics for runoff generation of boreal streams. In *Journal of Hydrology* 344 (3-4), pp. 198–209. DOI: 10.1016/j.jhydrol.2007.07.010.

Ritchie, Joe T. (1972): Model for predicting evaporation from a row crop with incomplete cover. In *Water Resour. Res.* 8 (5), pp. 1204–1213. DOI: 10.1029/WR008i005p01204.

Rodriguez, Nicolas B.; McGuire, Kevin J.; Klaus, Julian (2018): Time-Varying Storage - Water Age Relationships in a Catchment with a Mediterranean Climate. In *Water Resour Res*. DOI: 10.1029/2017WR021964.

Smith, Aaron A.; Tetzlaff, Doerthe; Soulsby, Chris (2018): Using StorAge Selection functions to quantify ecohydrological controls on the time-variant age of evapotranspiration, soil water, and recharge. In *Hydrol. Earth Syst. Sci. Discuss.*, pp. 1–25. DOI: 10.5194/hess-2018-57.

Sprenger, Matthias; Tetzlaff, Doerthe; Buttle, J. M.; Laudon, H.; Leistert, Hannes; Mitchell, Carl P. J. et al. (2018): Measuring and modelling stable isotopes of mobile and bulk soil water. In *Vadose Zone J*. DOI: 10.2136/VZJ2017.08.0149.

Stähli, Manfred; Nyberg, Lars; Mellander, Per-Erik; Jansson, Per-Erik; Bishop, Kevin 
[revised manuscript text omitted]